# Dot1 regulates nucleosome dynamics by its inherent histone chaperone activity in yeast

Soyun Lee[1], Seunghee Oh[1], Kwiwan Jeong[2], Hyelim Jo[1], Yoonjung Choi[1], Hogyu David Seo[1], Minhoo Kim[1], Joonho Choe[1], Chang Seob Kwon[3] & Daeyoup Lee[1]

Dot1 (disruptor of telomeric silencing-1, DOT1L in humans) is the only known enzyme responsible for histone H3 lysine 79 methylation (H3K79me) and is evolutionarily conserved in most eukaryotes. Yeast Dot1p lacks a SET domain and does not methylate free histones and thus may have different actions with respect to other histone methyltransferases. Here we show that Dot1p displays histone chaperone activity and regulates nucleosome dynamics via histone exchange in yeast. We show that a methylation-independent function of Dot1p is required for the cryptic transcription within transcribed regions seen following disruption of the Set2–Rpd3S pathway. Dot1p can assemble core histones to nucleosomes and facilitate ATP-dependent chromatin-remodeling activity through its nucleosome-binding domain, in vitro. Global analysis indicates that Dot1p appears to be particularly important for histone exchange and chromatin accessibility on the transcribed regions of long-length genes. Our findings collectively suggest that Dot1p-mediated histone chaperone activity controls nucleosome dynamics in transcribed regions.

[1] Department of Biological Sciences, Korea Advanced Institute of Science and Technology, Daejeon 34141, Republic of Korea. [2] Biocenter, Gyeonggi Business & Science Accelerator, Suwon, Gyeonggi-do 16229, Republic of Korea. [3] Department of Chemistry and Biology, Korea Science Academy of KAIST, Busan 47162, Republic of Korea. Correspondence and requests for materials should be addressed to D.L. (email: daeyoup@kaist.ac.kr)

During transcription elongation, the Set2–Rpd3S pathway has been shown to suppress cryptic transcription by maintaining a histone exchange balance on transcribed regions. The absence of the Set2–Rpd3S pathway makes chromatin more accessible to histone chaperones such as Asf1p, which results in the active assembly of histones and consequently an increases in histone exchange. These findings show that the balance of histone exchange at transcribed regions is crucial for genomic stability[1–5].

The Dot1p-mediated histone mark, histone H3 lysine 79 methylation (H3K79me), is also enriched in transcribed regions, suggesting that this modification and/or Dot1p may contribute to regulating chromatin dynamics. Dot1p is reported to be the sole nucleosomal H3K79 methyltransferase and is well conserved from yeast to humans. It catalyzes the mono-, di-, and tri-methylations of H3K79; these stable histone modifications undergo slow exchange through a nonprocessive (distributive) mechanism in which Dot1p detaches from the core histone after each round of methylation[6–10]. Previous studies have shown that H3K79me is enriched in the transcribed regions of genes[11], suggesting that Dot1p may function in transcription elongation. However, the function of Dot1p in transcription is still unclear. Although the involvement of Dot1p in transcription is not fully understood in yeast, some studies have shown that its mammalian ortholog, Dot1L, is involved in transcription and is essential for embryonic development in mammals[12, 13]. Human Dot1L (Dot1-Like histone methyltransferase) has been shown to bind RNAPII through its C-terminal domain and regulate the expression of actively transcribed genes[14]. Moreover, human Dot1L reportedly contributes to RNAPII-mediated elongation as part of the super elongation complex, which is involved in oncogenic transcriptional activation in mixed-lineage leukemia (MLL)[15–20]. Collectively, the existing evidence suggests that Dot1p is an evolutionarily well-conserved chromatin factor that can contribute to euchromatin through yet-unknown functions.

In this study, we demonstrate that yeast Dot1p participates in transcription regulation in a manner independent of its H3K79 methyltransferase activity. We used Set2p-depleted cells to assess the function of Dot1p in an environment of increased cryptic transcription and histone exchange. Based on the findings from our genome-wide, genetic, and biochemical analyses, we propose that Dot1p regulates the accessibility of chromatin through its histone-chaperoning and nucleosome-binding activities, and is involved in histone exchange.

## Results

**H4K16ac determines H3K79me localization by binding Dot1p.** The N-terminal basic patch of histone H4 interacts with a C-terminal acidic patch of Dot1p, which is required for the H4 tail binding of Dot1p and the Dot1p-mediated deposition of H3K79me[21, 22]. We hypothesized that an acetylation-mediated change in the charge of the H4 tail (K5, K8, K12, and K16) would modulate its interaction with Dot1p. To investigate the involvement of H4 acetylation in H3K79me deposition, we measured global H3K79me levels by western blot analysis in yeast strains in which each respective lysine residue had been mutated to alanine. Interestingly, the *H4K16A* mutation significantly decreased the global level of H3K79me3, whereas the other mutations had no apparent effect (Fig. 1a). H3K79me3 was also decreased in the non-acetylation mimic mutant, *H4K16R*, confirming the correlation between H3K79me3 and H4K16ac (histone H4 lysine 16 acetylation) (Supplementary Fig. 1a). As shown in Supplementary Fig. 1b, we found that H3K79me2/3 was decreased in *sas2Δ* mutant, a null mutant of H4K16 acetyltransferase Sas2p[23, 24], but not significantly altered in *esa1-414* cells, a catalytic temperature-

sensitive mutant of the NuA4 histone H4 acetyltransferase complex[25]. Thus, we conclude that Sas2p-mediated H4K16ac undergoes crosstalk with H3K79me.

To examine the nature of this crosstalk, we used chromatin immunoprecipitation sequencing (ChIP-seq) to analyze the genome-wide distributions of both modifications in wild-type and *sas2Δ* cells. We observed widespread differences in H3K79me3 and H3K79me1 between wild-type and *sas2Δ* cells, particularly in the distribution of H4K16ac on transcribed regions (Fig. 1b). Average plots and heatmaps representing our ChIP-seq data are shown in Fig. 1c–e. In wild-type cells, H3K79me3 was mainly distributed in the transcribed regions of genes. In *sas2Δ* cells, H3K79me3 gradually decreased towards the 3′-ends of transcribed regions, whereas the enrichment was largely unchanged near promoter regions (Fig. 1c). A genome-wide approach revealed a difference in the distribution pattern of H3K79 monomethylation (H3K79me1) that was not identified by our western blot analysis: H3K79me1 was anti-correlated with H3K79me3 in wild-type and *sas2Δ* cells. In wild-type cells, H3K79me1 was enriched on the 5′- and 3′-ends of transcribed regions but found at relatively low levels through the transcribed regions. In contrast, *sas2Δ* cells showed decreased enrichment at the 5′- and 3′-ends and increased H3K79me1 levels throughout transcribed regions (Fig. 1d). Calculation of H3K79me levels (reads per kilobase of transcript per million mapped reads (RPKM)) in transcribed regions with high H3K79me3 levels in wild-type cells revealed that H3K79me3 was decreased and H3K79me1 was increased in *sas2Δ* cells (Supplementary Fig. 1c, $n = 2672$). Unlike other histone methyltransferases, Dot1p is known to introduce multiple methyl groups via a nonprocessive (distributive) mechanism[10]. Thus, we hypothesized that the decrease of H3K79me1/3 level in *sas2Δ* mutant cells could reflect a reduction in the contact between Dot1p and chromatin. Supporting this idea, regions with high H3K79me1 and low H3K79me3 levels in wild-type cells showed even lower H3K79me1 enrichment in *sas2Δ* cells. Based on the nonprocessive activity of Dot1p, we inferred that H4K16ac could regulate the binding preference of Dot1p to transcribed regions (Supplementary Fig. 1d, e; $n = 1204$). We also observed that H4K16ac was broadly distributed on the transcribed regions of wild-type cells (Fig. 1b, bottom), and there was marked reduction in *sas2Δ* cells at transcribed regions but not near promoter regions (Fig. 1e). Concordant with our earlier observations (Fig. 1a), H3K79me3 was significantly reduced at transcribed regions with low H4K16ac levels in *sas2Δ* cells (Fig. 1c). These findings indicate that the relative enrichment of H3K79me3 at promoter regions in *sas2Δ* cells is likely due to the relatively high levels of H4K16ac. In the broader sense, our results suggest that the distribution of H4K16ac globally regulates H3K79me1 and H3K79me3 on transcribed regions. Although we attempted to perform a Dot1p ChIP-seq (or ChIP-quantitative PCR) analysis, we were unable to reproducibly identify regions of Dot1p occupancy, perhaps because Dot1p is expressed at a relatively low level and is distributed throughout the chromatin.

Based on the specific reduction of H3K79me3 levels in *sas2Δ* cells, we hypothesized that Dot1p specifically recognizes H4K16ac. To investigate the preferential binding of Dot1p to H4K16ac, we performed peptide pull-down assays using biotin-labeled N-terminal histone H4. Indeed, Dot1p repeatedly showed a binding preference for the H4K16ac peptide (Supplementary Fig. 1f; compare H4 to H4K16Ac), and this preference was confirmed by using the BLItz system (ForteBio, Inc.) (Supplementary Fig. 1g). Therefore, based on the distribution of H3K79me and the binding kinetics of Dot1p, we conclude that H4K16ac is important for the stable binding of Dot1p on chromatin.

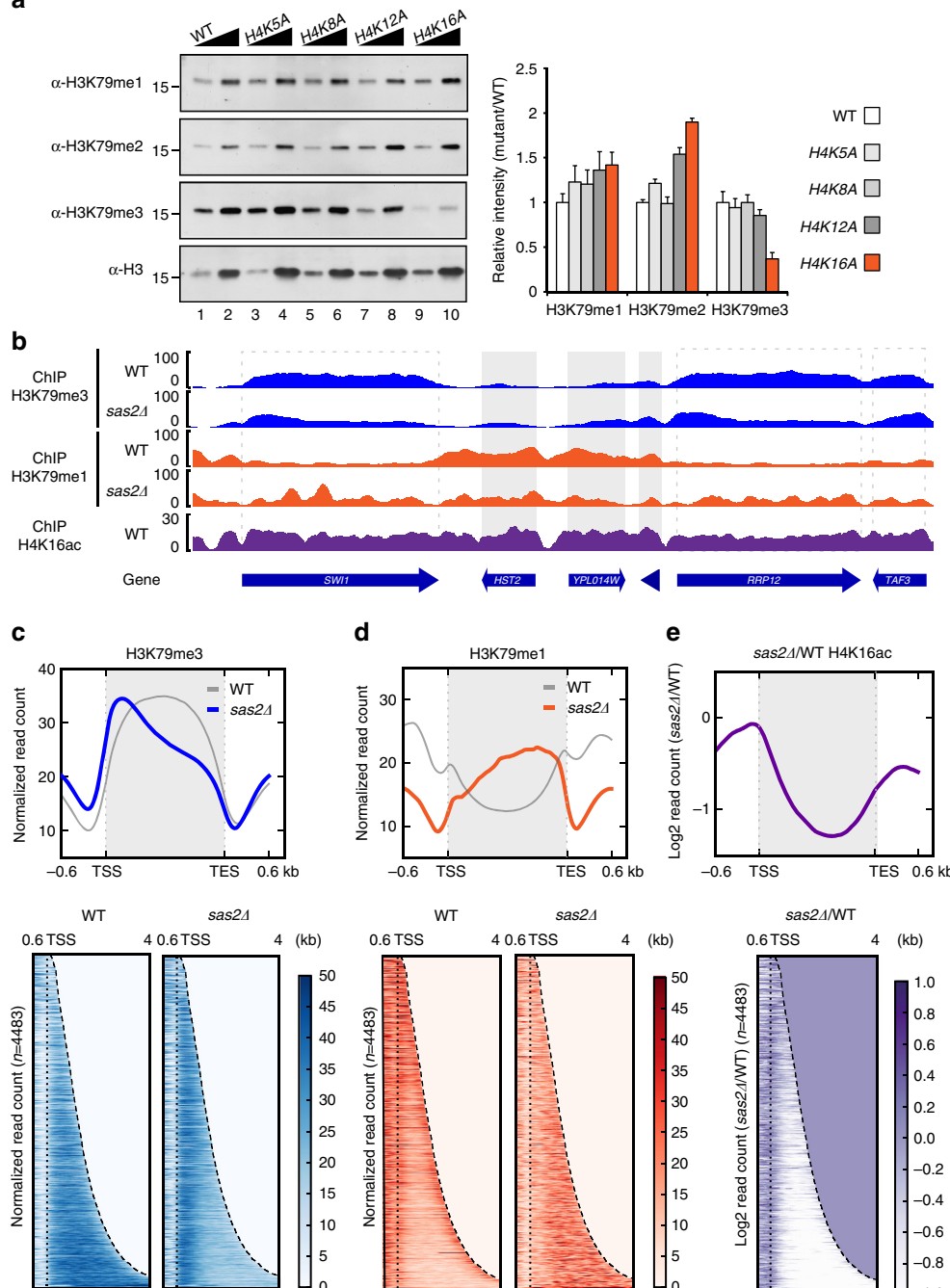

**Fig. 1** H4K16 acetylation regulates the Dot1p-mediated distribution of H3K79 methylation on euchromatin. **a** Western blot analysis for the levels of H3K79 mono-/di-/tri-methylation in histone lysine-to-alanine mutant strains. We used *wzy42* strains of which the sequences of *HHT1-HHF1* and *HHT2-HHF2* genes are deleted, and maintained by the inserted plasmid *pwz414-F13-HHT2-HHF2* to express histone H3 and H4. We modified the plasmid to express mutant histones H3 and H4. The right panel shows a quantification of the western blot data from three repeats. The error bars represent the s.d. for the biological replicates. **b** Distribution of H3K79me3 (blue) and H3K79me1 (orange) in wild-type and *sas2Δ* mutant cells. The values of the *y*-axis indicate normalized ChIP-seq read counts. The ChIP-seq data were obtained from biological duplicates. The distribution of H4K16ac in wild-type cells was indicated in purple. The dotted-line boxes indicate transcribed regions (*SWI1*, *RRP12*, and *TAF3*), while the gray boxes (*HST2*, *CIP1*, and *MRPS16* (indicated as a triangle with no gene name label)) mark regions with low H3K79me3 levels. **c**, **d** Heatmaps and average plots showing the distributions of H3K79me3 (**c**, blue) and H3K79me1 (**d**, orange) in wild-type and *sas2Δ* mutant cells. **e** The H4K16ac level calculated as the log$_2$ fold change in *sas2Δ* cells over wild-type cells (purple). The average plot and heatmap indicate differential distribution of H4K16ac on transcribed regions. Profiles are sorted by ascending length of their transcribed regions. **c–e** The genes were sorted in descending order of their H3K79me3 levels on transcribed regions in *sas2Δ* cells (*n* = 4483). The read count data were obtained from biological duplicates. In the average plots, the values of the *y*-axis indicate normalized ChIP-seq read counts, and the *x*-axis indicate distance from transcription start sites (TSS) and transcription end sites (TES). For the heatmaps, the *y*-axis indicates each genes and the *x*-axis indicates distance from TSS. The genes in heatmap was arranged in order of length. The intensity of color indicates the value of normalized read counts from 0 to 50. The dotted lines indicate the TSS and TES. All values were normalized to the read count, and the value of the region exceeding the TES was treated as 0

**Dot1p participates in incorporating new histones**. Since H4K16ac is known to decondense chromatin and hence increases the accessibility of chromatin which facilitates transcription[26], together with our finding of preferential recruitment of Dot1p to regions with high levels of H4K16ac (Fig. 1e), we speculated that Dot1p may be involved in transcription elongation. Because the Set2–Rpd3S pathway is also involved in transcription elongation to suppress cryptic transcription, we examined the effects of Dot1p on cryptic transcription via Set2p.

We tested the extent of cryptic transcription of Dot1p by performing spotting assays using *pGAL1-FLO8-HIS3*-inserted

strains[27]. Wild-type cells displayed weak growth due to the absence of *HIS3* gene expression, whereas *set2Δ* cells showed robust growth due to increased cryptic transcription of the *HIS3* gene (Supplementary Fig. 2a). Interestingly, a significant growth defect was observed in the *dot1Δset2Δ* mutant, suggesting that Dot1p modulates this cryptic transcription. However, *sas2Δset2Δ* cells displayed no growth defect compared to *set2Δ* cells, indicating that Sas2p may play a less pronounced role in this cryptic transcription. These results collectively suggest that Dot1p may suppress the cryptic transcription, while the Sas2-mediated H4K16ac plays only a marginal role to suppress cryptic transcription.

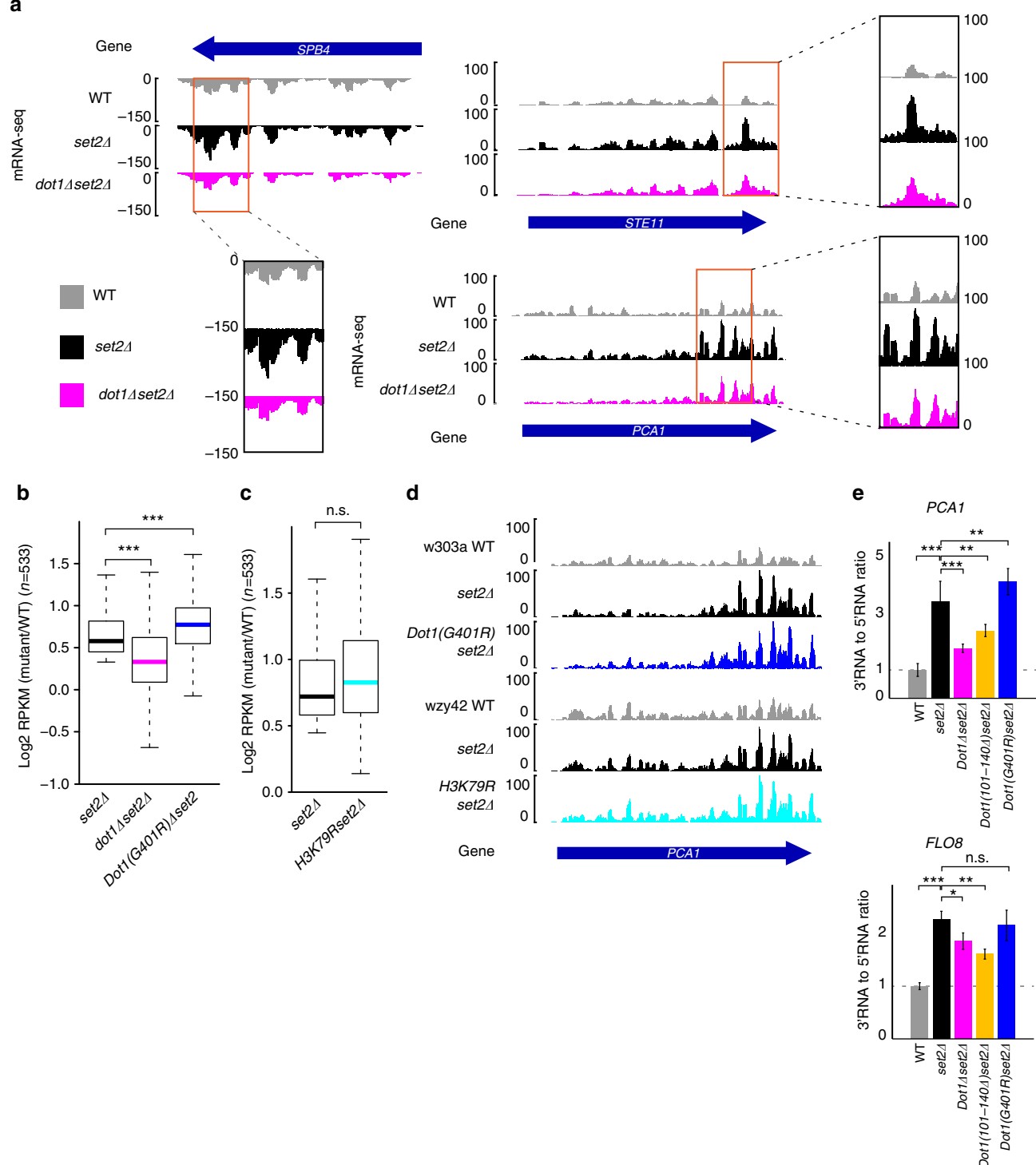

To investigate the involvement of Dot1p in cryptic transcription via the Set2–Rpd3S pathway, we performed mRNA-seq analyses in *set2Δ* and *dot1Δset2Δ* mutants. As expected, the increased mRNA expression observed in *set2Δ* cells was suppressed in *dot1Δset2Δ* cells (Fig. 2a; compare *set2Δ* and *dot1Δset2Δ*), while there was no change in mRNA levels in *sas2Δ* and *dot1Δ* mutants compared to wild type (Supplementary Fig 2b). To assess the genome-wide changes in mRNA expression levels, we classified gene groups with significantly increased mRNA levels in *set2Δ* mutants compared to wild-type cells ($n = 533$). Analysis of $\log_2$ fold changes revealed that this gene set was significantly suppressed in *dot1Δset2Δ* cells compared to *set2Δ* cells (Fig. 2b). Consistent with a previous report that a group of cryptic promoters ($n = 213$) showed decreased expression in *dot1Δset2Δ* versus *set2Δ* mutants[28], we confirmed deletion of *DOT1* suppressed cryptic transcription (Supplementary Fig. 2c). These findings suggest that Dot1p contributes to spurious transcription from cryptic promoters under Set2p-depleted conditions.

Previous reports showed that the increased mRNA expression level in *set2Δ* cells[4] is suppressed in *asf1Δset2Δ* cells[1], as the loss of the histone chaperone, Asf1p, prevents an uncontrolled histone exchange observed in *set2Δ* cells[1]. Here, we hypothesized that Dot1p may suppress cryptic transcription using a mechanism similar to that mediated by Asf1p. We thus compared the mRNA expressions of *asf1Δset2Δ* and *dot1Δset2Δ* mutants, to examine whether Dot1p and Asf1p overlap in repressing cryptic transcription. Consistent with the previous report[1], the mRNA level clearly decreased in *asf1Δset2Δ* cells, and slightly lower than in *dot1Δset2Δ* cells (Supplementary Fig. 2d). We therefore suggest that Asf1p facilitates the cryptic transcription more potently in *set2Δ* cells compared to Dot1p. To determine whether the genes suppressed in *dot1Δset2Δ* cells differed from those suppressed in *asf1Δset2Δ* cells, we compared gene sets with decreased mRNA levels in *asf1Δset2Δ* and *dot1Δset2Δ* mutants on the genes that were up-regulated in *set2Δ* cells ($n = 533$ genes). We found that the two gene groups did not overlap significantly (Supplementary Fig. 2e, upper panel). The same non-overlap was also observed in the gene sets with decreased mRNA levels (Supplementary Fig. 2e, bottom panel). These data suggest that Dot1p facilitates transcription separately from Asf1p.

The most well-known biochemical function of Dot1p is the methylation of histone H3K79[6–8]. To investigate whether the methyltransferase activity of Dot1p is involved in the observed transcription repression phenotype, we performed mRNA-seq in *set2Δ* cells harboring the methylation-defective mutation, *Dot1(G401R)*[8, 10], and the non-methylatable histone mutant, *H3K79R*. Surprisingly, neither *H3K79R* nor *Dot1(G401R)* repressed mRNA

expression in the *set2Δ* background (Fig. 2b, c). We examined mRNA expression patterns in specific cryptic promoter genes and found no distinguishable difference among wild-type, *Dot1 (G401R)set2Δ*, and *H3K79Rset2Δ* cells (Fig. 2d, *PCA1* gene). From these data, we conclude that Dot1p regulates cryptic transcription independently of its methyltransferase activity.

Lastly, we verified the observed suppression phenotype by performing quantitative reverse-transcription PCR for two representative cryptic genes, *PCA1* and *FLO8*. We found that the relative RNA ratio was decreased in *set2Δdot1Δ* cells compared to *set2Δ* cells, and no significant decrease was observed in *Dot1(G401R)set2Δ* cells compared to *set2Δ* cells (Fig. 2f), although the introduction of the nucleosomal binding-defective mutant *Dot1(101–140Δ)*[29] showed repression similar to that of *dot1Δset2Δ* cells (Fig. 2e). Taken together, these data indicate that the nucleosome-binding activity of Dot1p plays a pivotal role in its ability to suppress the cryptic transcription induced by the depletion of *SET2*.

**Loss of *DOT1* suppresses histone exchange in transcribed genes**. The up-regulation of transcription, including cryptic transcription, is reportedly associated with altered histone exchange[1]. Consistent with this, our mRNA-seq and ChIP-seq data showed that 85% of the genes that showed cryptic transcription in *set2Δ* cells also exhibited increases in histone exchange (Supplementary Fig. 3a). To examine whether Dot1p was involved in this alteration of histone exchange, we performed ChIP-seq on H4 acetylation (H4ac), a histone modification that shows a high correlation with histone turnover[1]. Our results revealed that the loss of *SET2* resulted in the accumulation of H4ac (normalized with respect to H3) over transcribed gene regions, whereas *set2Δdot1Δ* cells exhibited lower H4ac levels across these regions relative to the levels found in *set2Δ* cells (Fig. 3a). These data indicate that Dot1p is involved not only in cryptic transcription, but also in histone exchange during transcription.

Since the loss of *DOT1* seemed to affect the histone exchange rate in *set2Δ* cells, we used ChIP-seq of histone H3 to assess changes of H3 levels in transcribed regions. In budding yeast, the Set2–Rpd3S pathway has been reported to influence infrequently transcribed and/or relatively long genes, and the loss of *SET2* was associated with an increase of H3 in transcribed regions[1, 4]. For two representative long genes, *BLM1* (6.4 kb) and *POL2* (6.7 kb), we found that H3 was increased in *set2Δ* cells relative to wild-type cells, and that the loss of *DOT1* suppressed the increase of H3 in the *set2Δ* background (Fig. 3b; compare *set2Δ* and *dot1Δset2Δ* in the shadowed boxes). Furthermore, we found that the levels of H3

**Fig. 2** Loss of *DOT1* rescues the cryptic transcription of *set2Δ* cells in a methylation-independent manner. **a** mRNA-seq data in wild-type (gray), *set2Δ* (black), and *dot1Δset2Δ* (magenta) mutants for the cryptic loci, *SPB4*, *STE11*, and *PCA1*. The plus and minus values of the *y*-axis indicate level of mRNA read counts and the sense and antisense strands, respectively (left to right, +; right to left, −). The read count data were obtained from biological duplicates. The orange boxes indicate an enlarged view for peak of cryptic transcription in each loci. The values of zoom-up box represents *y*-axis values. **b** A boxplot of the $\log_2$ fold change values of *set2Δ* (black), *dot1Δset2Δ* (magenta), and Dot1(G401R)set2Δ (blue) mutants versus the levels in wild-type cells for genes that showed increased mRNA expression in *set2Δ* cells ($n = 533$). The Cuffdiff analysis program was used. The genes that passed the correction test in Cuffdiff were collected, and their fold change values versus wild-type levels were calculated. Data were obtained from biological duplicates. ***P*-value <0.001. **c** A boxplot of the $\log_2$ fold change values for the *set2Δ* (black) and *H3K79Rset2Δ* (turquoise) histone residue modified mutants versus their wild-type strain (*wzy42*). Data were obtained from biological duplicates; n.s., *p*-value >0.05 (Wilcoxon and Mann–Whitney tests). **d** mRNA-seq data were obtained for Dot1 methylation activity-associated mutants using the cryptic gene, *PCA1*. *set2Δ* cells (color black) were used as a positive control. *Dot1(G401R)set2Δ* was based on the *w303a* wild-type (color blue, upper windows), while *H3K79Rset2Δ* was based on the *wzy42* wild-type (color turquoise, lower windows). The values of the *y*-axis indicate level of mRNA read counts. **e** RT-qPCR analysis of cryptic transcription at the cryptic loci, *FLO8* (upper) and *PCA1* (bottom) from three biological repeats. The expression level of the 3′-end of each RNA was normalized by that of the corresponding 5′-end. The mutant values were normalized by that of the wild-type. The error bars represent the s.d. for the biological replicates. ***P*-value <0.001; **p*-value <0.01; *p*-value <0.05; n.s., *p*-value >0.05 generated by analysis of variance (ANOVA) using the R statistic program. The levels of mRNA were quantified by quantitative PCR using the primers listed in Supplementary Table 3

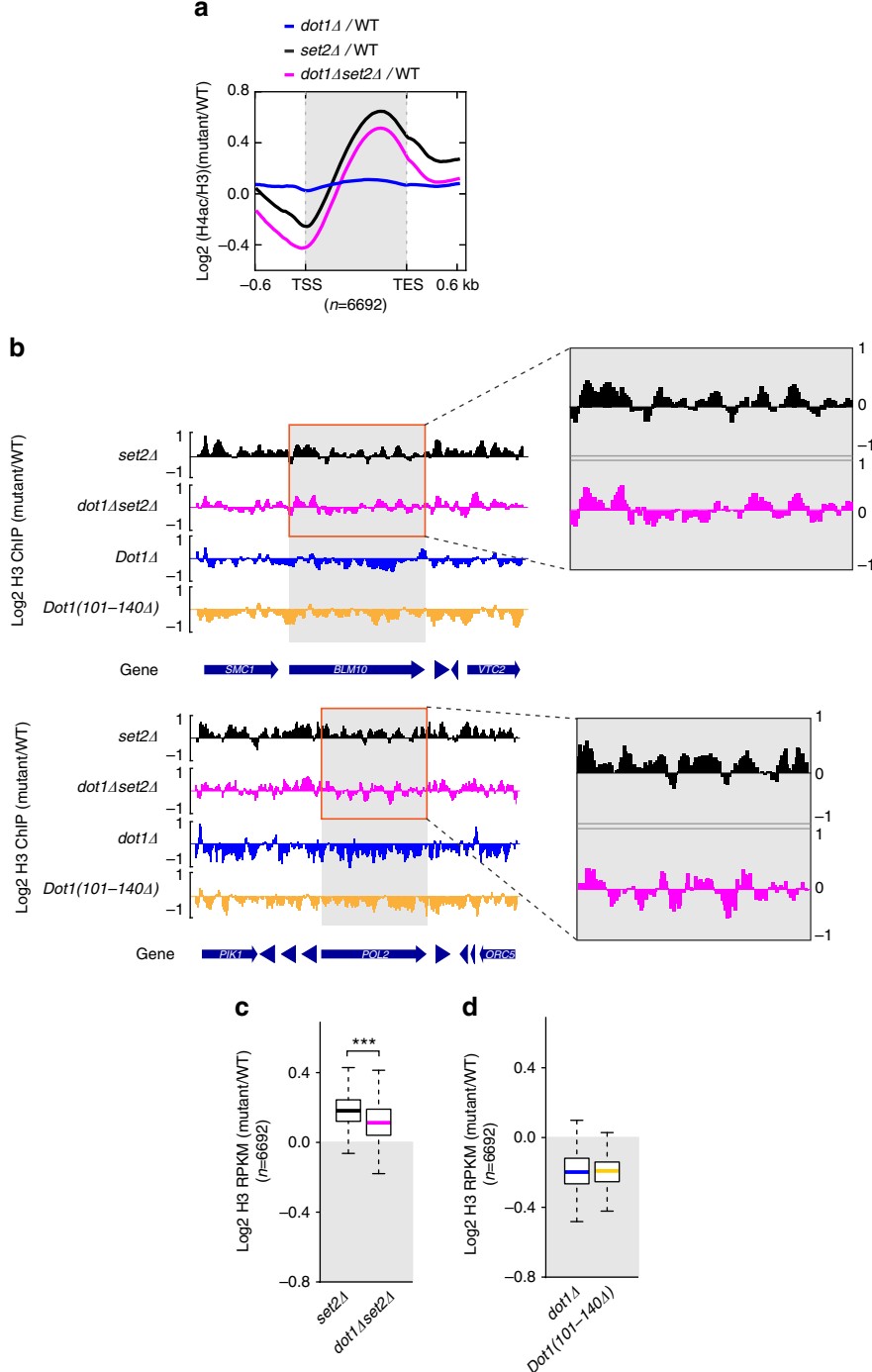

**Fig. 3** Loss of *DOT1* suppresses the increased histone exchange at transcribed regions in *set2Δ* cells. **a** The ChIP-seq average plot of H4ac normalized by H3. The values for *set2Δ* (black), *dot1Δset2Δ* (magenta), and *dot1Δ* (blue) cells were calculated as the log$_2$ ratios of the mutant over wild type. The values of the *y*-axis indicate log$_2$ fold change level of H4ac/H3 ChIP-seq in mutants over wild-type cells, and the *x*-axis indicates the distance from transcription start sites (TSS) and transcription end sites (TES). The data set included all transcribed regions of yeast (number of genes, *n* = 6692), and were obtained from biological duplicates. **b** Distribution of H3 at the *BLM10* (left) and *POL2* (right) genes. The ChIP-seq data were obtained from biological duplicates. The *y*-axis indicates log$_2$ fold change of H3 in mutants over wild type. The H3 levels were normalized as log$_2$ (mutant/wild-type) values using the spike-in normalization method. The plus and minus values of the *y*-axis indicate the increased and decreased level of each mutant with respect to wild type. The gray boxes indicate the transcribed regions of *BLM10* and *POL2*. The orange boxes indicate an enlarged view for *BLM10* and *POL2* loci. The *y*-axis values represent the values of zoom-up box. **c** A boxplot of histone H3 log$_2$ fold change values (normalized by the spike-in method). The *y*-axis indicates log$_2$ fold change of H3 in mutants over wild type. The values presented for *set2Δ* (black) and *dot1Δset2Δ* (magenta) are relative to the wild-type values at the transcribed regions of all yeast genes (number of genes, *n* = 6692). The shadowed box indicates negative values, indicating the decreased region in mutant with respect to wild-type. ***P-value <0.001 (Wilcoxon and Mann–Whitney tests). **d** A boxplot of histone H3 log$_2$ fold change values obtained for *dot1Δ* (blue) and *Dot1(101–140Δ)* (yellow) at the transcribed regions of all yeast genes (number of genes, *n* = 6692). The *y*-axis indicates log$_2$ fold change of H3 in mutants over wild type. The shadowed box indicates negative values, indicating the decreased region in mutant with respect to wild type

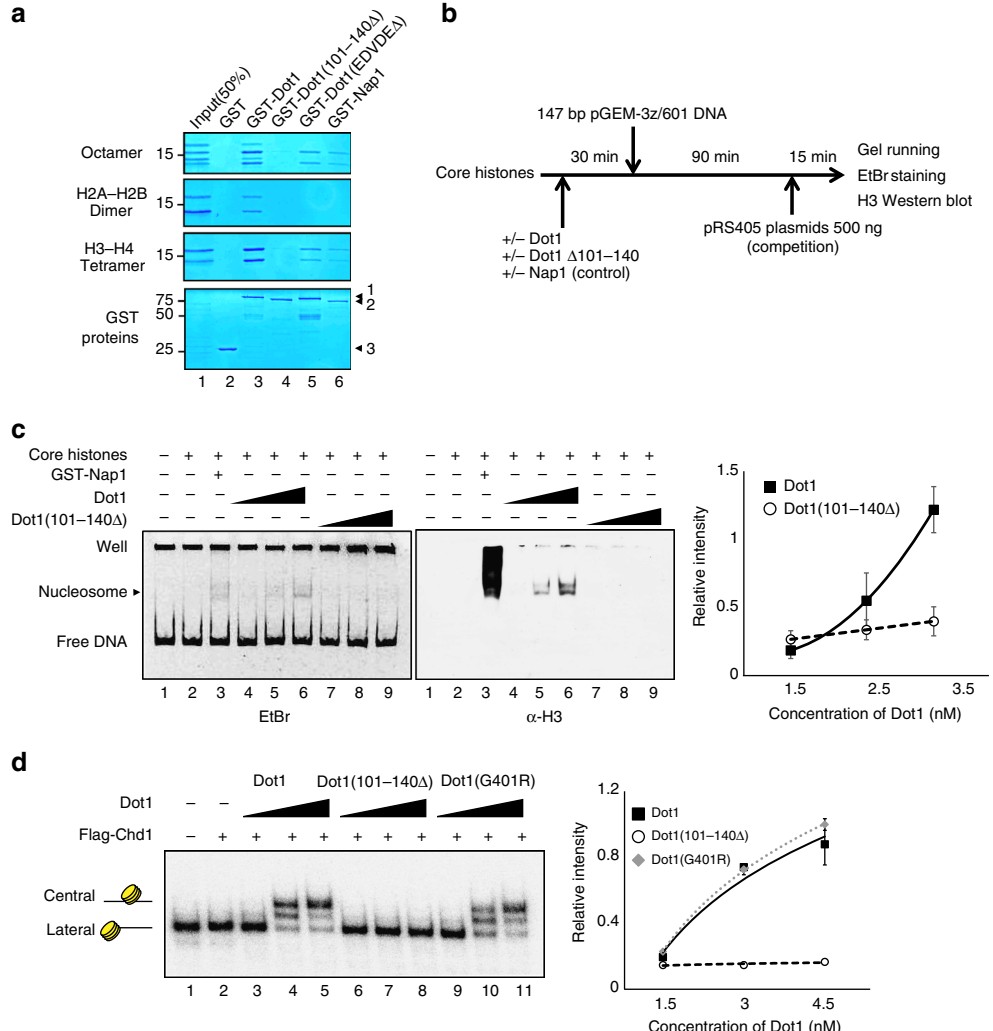

**Fig. 4** Dot1p has nucleosome assembly activity and enhances chromatin remodeler activity. **a** Protein pull-down assay of GST-tagged Dot1 proteins against core histones. SDS-acrylamide gels were stained with Coomassie blue. The Octamer panel (top) indicates histone H2A, H2B, H3, and H4 for binding with GST (negative control), GST-Dot1, a nucleosome binding-defective mutant GST-Dot1 (101–140Δ), a histone H4 tail binding-defective mutant GST-Dot1 (EDVDEΔ), and GST-Nap1 proteins. The H2A–H2B dimer (second panel) represents band size of histone H2A–H2B for each protein binding. The H3–H4 tetramer panel (third panel) indicates size of histone H3–H4 for each protein binding. The 5 μg of GST fusion proteins were incubated with 10 μg of histones in 200 μl of PDB buffer for 2 h at 4 °C. In the bottom panel, arrows indicate the following: 1, GST-Dot1p, GST-Dot1p (EDVDEΔ), and GST-Dot1p (101–140Δ) (showing a smaller size); 2, GST-Nap1p; and 3, GST. **b** Experimental scheme of the in vitro nucleosome assembly assay. **c** Assay of the ability of Dot1 to assemble nucleosomes on a fragment of pGEM-3z/601 DNA. The 147 bp DNA fragment (100 ng) was incubated without (lane 1) or with (lanes 2–9) 1:1 mass ratios of core histones. The nucleosome assembly activities of wild-type Dot1 (1.5, 2.4, and 3.2 nM; lanes 4–6) and the nucleosome binding-defective mutant, Dot1(101–140Δ) (same concentrations as Dot1; lanes 7–9) are shown, with 2 μg of GST-Nap1 used as a positive control for nucleosome assembly (lane 3). The arrow labeled with 'nucleosome' indicates size of assembled nucleosome. The nucleosome assembly was visualized with ethidium bromide (EtBr) staining (left) and histone H3 western blotting (center), and the data were quantified (right). The data are presented are averages of three experiments, and error bars represent the s.d. for the biological replicates. **d** The ability of Dot1 to stimulate chromatin remodeling does not require its methyltransferase activity. Sliding assays using the nucleosome remodeler, Chd1p, were performed with Dot1p (lanes 3–5), nucleosome binding-defective Dot1(101–140Δ) (lanes 6–8), and methyltransferase activity-defective mutant Dot1 (G401R) (lanes 9–11). The graph (right) indicates the relative intensities of the central nucleosome bands versus the Dot1 protein concentration. The data presented are the averages of three experiments, and error bars represent the s.d. for the biological replicates

in *dot1Δ* and *Dot1(101–140Δ)* cells were lower than those in wild-type cells. Next, we analyzed the average log$_2$ fold change values of H3 levels on transcribed regions for each mutant versus the wild type. We found that the H3 level was slightly higher in *set2Δ* cells compared to wild-type cells, whereas the level in *dot1Δset2Δ* cells was lower than that in *set2Δ* cells (Fig. 3c). The average levels of H3 were decreased in the *dot1Δ* and *Dot1(101–140Δ)* mutants, which showed suppression of the cryptic transcription triggered by the absence of *SET2* (Fig. 3d). These results suggest that the

depletion of *DOT1* decreases the level of H3 through the nucleosome-binding function of Dot1p. A previous report found that depletion of the histone chaperone, Asf1p, in the *set2Δ* background decreased the histone exchange rate, even though histone exchange was not significantly altered in *asf1Δ* cells[1]. Furthermore, the ChIP enrichment of H3 was reportedly diminished on the transcribed region of genes in *asf1Δ* cells[30]. The level of H4ac did not show a significant difference in *sas2Δset2Δ* cells compared to *set2Δ* cells (Supplementary Fig. 3b).

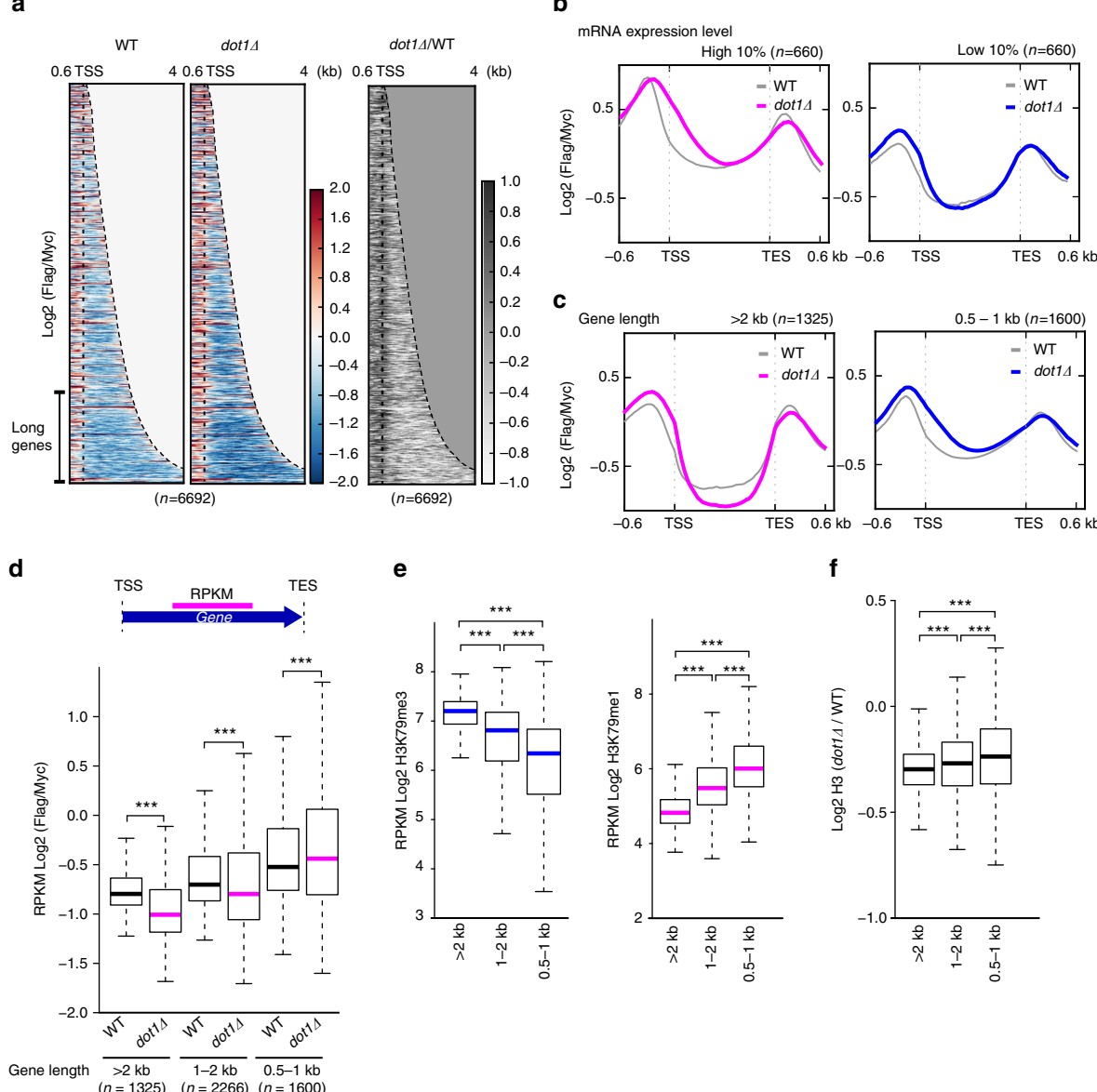

**Fig. 5** Dot1p preferentially affects histone exchange in long genes. **a** Heatmap of histone exchange (log₂ Flag ChIP/Myc ChIP) at all genes of yeast (n = 6692). The read counts were obtained from biological duplicates. Profiles are shown in ascending order of the length of the transcribed region. Dotted lines indicate the TSS. Genes >2 kb in length are indicated as 'Long Genes'. The value of the region exceeding the TES is treated as 0. Left, wild-type; center, dot1Δ; right, dot1Δ relative to wild type. The color bar represents value of log₂ Flag/Myc (color red: >0, color blue: <0, for left and center panel; color black: >0, color white: <0, for right panel). **b**, **c** Total genes were clustered based on their **b** mRNA expression levels and **c** gene lengths. The gray line indicates the histone exchange rate in wild-type cells. Magenta lines indicate clusters of genes having high mRNA expression levels (**b**, n = 660) and those longer than 2 kb in length (**c**, n = 1325) in dot1Δ cells. Blue lines indicate clusters of genes having moderate mRNA expression levels (**b**, n = 660) and those of 0.5–1 kb in length (**c**, n = 1600). The values of the y-axis indicate histone exchange, and the x-axis indicate distance from TSS and TES. **d**, **e** ***P-value <0.001. The upper panel of (**d**) shows a schematic diagram of our analysis. Magenta colored line, the center of transcribed regions, represents the region where taken to be values in the boxplots in (**d**, **e**). **d** A boxplot of histone turnover RPKM values obtained at the centers of the transcribed regions in wild-type and dot1Δ cells for each gene length cluster (i.e., >2 kb, 1–2 kb, and 0.5–1 kb). **e** A boxplot of the RPKM values for H3K79me3 and H3K79me1 at the centers of transcribed regions. Genes are clustered based on gene length as described for (**d**), and the boxplots present H3K79me3 (left) and H3K79me1 (right). **f** The level of histone H3 is clearly related to gene length. A boxplot presents the log₂ fold change of H3 in dot1Δ versus wild-type cells at the centers of transcribed regions, as normalized by gene length (RPKM)

This may be due to the difference in the chromatin context in wild-type and set2Δ cells. When SET2 is lost, the chromatins is hyper-acetylated as the histone deacetylase enzymes are unable to be recruited via Set2–Rpd3S pathway. The level of H3 in sas2Δ was lower than wild-type cells (Supplementary Fig. 3c), and histone turnover measured by Flag-histone H3 in an inducible strain displayed a slight decrease in sas2Δ (Supplementary

Fig. 3d). These results support the hypothesis that the Dot1p recruitment by Sas2p-mediated H4K16ac on transcribed region regulates histone exchange. Taken together, the chromatin state in set2Δ cells is different from that of wild type, such that Sas2p-mediated H4K16ac may no longer be necessary for the recruitment of Dot1p. Based on the present and previous

findings, we hypothesize that Dot1p is likely to be linked to the assembly of histones during transcription.

**Dot1p has a nucleosome assembly activity in vitro.** We previously identified the nucleosome-binding domain within Dot1p[29]. To further investigate the nucleosome-binding nature of Dot1p, we performed protein pull-down assays against histone octamers, H2A–H2B dimers, and H3–H4 tetramers (Fig. 4a, lane 3). Similar to the histone chaperone, Nap1p (positive control)[31], Dot1p bound to all of the tested histone combinations (Fig. 4a, compare lanes 3 and 6). Compared to wild-type Dot1p, the H4 tail binding-defective mutant Dot1 (EDVDE$\Delta$) mutant protein showed a marked decrease in the binding of H2A–H2B dimers and a smaller decrease in the binding of H3–H4 tetramers (Fig. 4a, lane 5). Dot1p (101–140$\Delta$) mutant proteins showed clear decreases in the bindings of H2B–H2A dimers and H3–H4 tetramers (Fig. 4a, lane 4). These data indicate that the nucleosome-binding domain of Dot1p is important for its association with core histones. Taken together, these results support the possibility that Dot1p might have nucleosome-assembling activity in situations where core histones assemble to form nucleosomes.

To confirm this hypothesis, we carried out an in vitro nucleosome assembly assay (for details, see Fig. 4b). In the presence of Dot1p, we found that core histones were assembled into nucleosomes, as assessed by ethidium bromide (EtBr) gel staining and electrophoretic mobility shift assay/western blot analysis with an antibody against H3 (Fig. 4c, lanes 4–6). However, no nucleosome assembly activity was observed in Dot1 (101–140$\Delta$) cells (lanes 7–9). When the quantified relative intensity values of three independent experiments were plotted against the Dot1p concentration, we confirmed the nucleosome assembly activity of Dot1 was dependent on nucleosome-binding domain (101–140) (Fig. 4c, right panel). Together, these experiments indicate that Dot1p is able to assemble nucleosomes in a manner similar to that of other histone chaperones, such as Nap1p and Asf1p[32].

Histone chaperones (e.g., Asf1p) are known to have both nucleosome assembly and disassembly activities[32–35], and to promote adenosine triphosphate (ATP)-dependent remodeling in vitro and in vivo[36–38]. Since our results suggest that Dot1p is likely to be a histone chaperone that is involved in histone exchange and cryptic transcription, we hypothesized that Dot1p may stimulate ATP-dependent chromatin remodelers in vitro. To test this hypothesis, we performed nucleosome-sliding assays using Chd1p (a chromatin remodeler known to remodel nucleosomes from the lateral to central positions)[39] and 601 positioned mononucleosomes in the presence of wild-type or mutant Dot1p (Fig. 4d). We found that wild-type Dot1p stimulated the nucleosome-sliding activity of Chd1p (Fig. 4d, lanes 3–5), whereas Dot1(101–140$\Delta$) did not (Fig. 4d, lanes 6–8). To exclude the possibility that the histone methyltransferase activity of Dot1p could be involved in chromatin remodeling, we performed sliding assays using the previously described catalytically inactive mutant, Dot1(G401R)[8, 10]. In contrast to Dot1 (101–140$\Delta$), Dot1(G401R) showed no defect in the ability to stimulate chromatin remodeling (Fig. 4d, lanes 9–11). Therefore, the nucleosome-binding activity of Dot1p, but not its histone methyltransferase activity, is essential for its ability to stimulate chromatin remodeling in vitro. To test whether this stimulation is restricted to Chd1p, we performed sliding assays with the RSC (Remodeling the Structure of Chromatin) and Isw1 complexes (Supplementary Fig. 4). The RSC complex is known to slide nucleosomes from the central to lateral position, whereas Chd1p and Isw1p are reported to have the opposite sliding

activity[40, 41]. We found that Dot1p stimulated the nucleosome-sliding activities of all tested remodeling complexes in a nucleosome binding-dependent manner (Supplementary Fig. 4, lanes 5 and 6). These findings suggest that Dot1p does not directly affect the activity of chromatin remodelers per se, but rather engages the nucleosome to create a more suitable environment for chromatin remodeling.

**Dot1p preferentially affects histone exchange in long genes.** Based on our finding that the *dot1$\Delta$* mutation rescued the cryptic transcription in *set2$\Delta$* cells, we set out to characterize the function of wild-type Dot1p in histone exchange. To assess replication-independent histone exchange in *dot1$\Delta$* cells, we used an in vivo histone exchange system[1, 42]. Analysis of the whole gene set showed that histone turnover was disrupted in *dot1$\Delta$* cells relative to wild-type cells (Fig. 5a). In wild-type cells, histone exchange was mainly activated (red color) upstream of transcription start sites (TSSs); in *dot1$\Delta$* cells, in contrast, histone turnover was increased near the promoter regions (5'-ends of genes). Notably, the histone turnover of *dot1$\Delta$* cells was more dramatically decreased (blue color) relative to that in wild-type cells when we considered genes with long transcribed regions (Fig. 5a, indicated as the 'Long genes' range). A heatmap presenting the relative ratio of histone turnover in *dot1$\Delta$* over wild-type cells (Fig. 5a, right panel) showed that histone turnover was decreased at transcribed regions (white color). These experiments demonstrate that Dot1p is globally involved in histone exchange.

We next examined whether the loss of *DOT1* affects specific gene groups divided by their transcription rates and gene lengths (Fig. 5b, c). Genes were grouped in order of their transcription rate, as assessed by mRNA-seq analysis. As shown in the average plot presented in Fig. 5b, the histone turnover of genes in *dot1$\Delta$* cells was increased from the TSS to the 5'-end, and the degrees of increase were similar between genes of high and moderate transcription rates. Interestingly, when genes were classified according to gene length, there was an obvious difference in the histone exchange of long (>2 kb) versus short (0.5–1 kb) genes in *dot1$\Delta$* cells (Fig. 5c). The average plot of the short genes resembled that of the transcription rate (Fig. 5b), with increased histone turnover observed near the TSS (Fig. 5c, right panel). For the long gene group, in contrast, the histone turnover was decreased at the centers of transcribed regions in *dot1$\Delta$* cells compared to wild-type cells (Fig. 5c, left panel). Thus, in *dot1$\Delta$* cells, the histone exchange rate near the TSS was generally high, whereas that at the center of transcribed regions showed a gene length-dependent decrease. Previous reports found that the increases of histone exchange and H3K56 acetylation in the *set2$\Delta$* mutant were more evident on longer-length genes[1, 4]. Since Dot1p appears to cooperate with the Set2p–Rpd3S pathway to repress cryptic transcription (Fig. 2), we focused on the role of Dot1p at transcribed regions by further analyzing its function with respect to gene length. To eliminate analysis error due to variability of gene length and the effect of strong histone exchange near the promoters and 5'-ends of genes, we calculated the RPKM of histone turnover at the centers of the transcribed regions of genes, excluding the 5'- and 3'-ends (Fig. 5d, upper panel). Interestingly, in long genes (>2 kb), the histone exchange rate was significantly decreased in *dot1$\Delta$* versus wild-type cells ($n$ = 1325). This result reflects the average plot shown in the graph of Fig. 5c. Histone turnover was also decreased, but to a lesser degree, in *dot1$\Delta$* mutant cells for genes of 1–2 kb in length ($n$ = 2266); thus, it seems that histone exchange decreases with the distance from the TSS, at least in this case. For genes of 0.5–1 kb in length, the plot showed that the histone exchange of the transcribed region was slightly increased in *dot1$\Delta$* cells ($n$ = 1600).

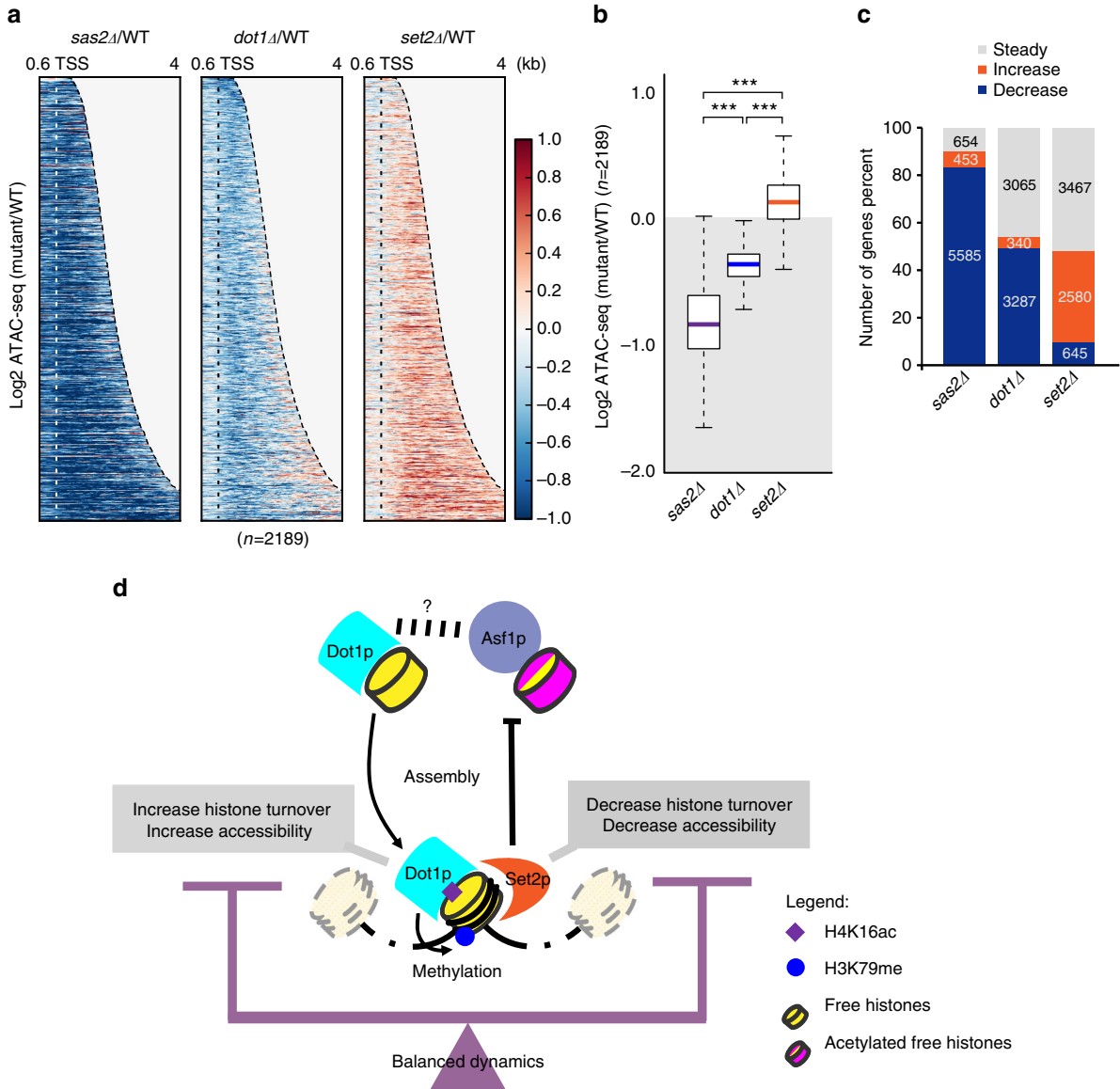

**Fig. 6** Dot1p facilitates nucleosome accessibility in the transcribed region of genes. **a** Heatmap for the log$_2$ fold change of the ATAC-seq signal in sas2Δ (left), dot1Δ (center), and set2Δ (right) cells versus wild type. Increased accessibility is indicated in red, and decreased accessibility is indicated in blue. Genes were ordered by their accessibility difference in dot1Δ versus wild-type cells (n = 2189). Data were obtained from biological duplicates. The y-axis indicates each genes and the x-axis indicates distance from TSS. The genes in heatmap was arranged in order of gene length. The color bar represents value of log$_2$ mutant compared to wild-type (color red: >0, color blue: <0). The dotted lines indicate the TSS. All values were normalized to the read count, and the value of the region exceeding the TES was treated as 0. **b** A boxplot of ATAC signal log$_2$ fold changes in mutants over wild type. The value of ATAC signal implies chromatin accessibility. The values of the y-axis indicate log2 fold change level of ATAC-seq value in each sas2Δ (purple), dot1Δ (blue), and set2Δ (orange) mutant over wild-type cells. ***P-value <0.001, calculated by Wilcoxon and Mann–Whitney tests, and the shadowed box represents negative values. **c** Stacked percent graph for the difference in accessibility between mutant and wild-type cells. Log$_2$ fold changes between mutants (sas2Δ, dot1Δ, and set2Δ) and wild-type on transcribed regions were analyzed using Bwtool and the following classification criteria: increased (orange), log$_2$ fold change >0.2; decrease (blue), log$_2$ fold change <−0.2; steady (gray), −0.2 <log$_2$ fold change <0.2. **d** The proposed model. Dot1p binds the nucleosome with the support of H4K16ac (purple diamond), and increases histone turnover and accessibility via its inherent histone chaperone activity (nucleosome binding). Conversely, Set2 acts to decrease histone turnover and accessibility, thereby balancing nucleosome dynamics. Dot1p methylates H3K79 by nucleosome binding (blue circle), but this activity is independent of its involvement in balancing nucleosome dynamics

Therefore, we conclude that Dot1p contributes to histone exchange in a gene length-dependent manner (Fig. 5d).

We hypothesized that strong RNAPII-mediated histone exchange near the promoter could affect the gene length-related difference in the histone exchange rate of dot1Δ cells. To test the possibility of the short genes being closer to the promoter where RNAPII is enriched highly, we analyzed the average level of RNAPII (anti-8WG16) along the gene, and compared these values between the gene length groups. As shown in Fig. S5, RNAPII was highly enriched at promoters and across the 0.5–1 kb genes, but reduced across the long genes (>2 kb). Therefore, it is possible that short genes (0.5–1 kb) are more likely to be affected by strong RNAPII-mediated histone exchange, although this mechanism may not be the sole contributor.

To assess whether Dot1p affects the histone exchange rate on long genes (>2 kb), we examined the levels of H3K79me3 and

H3K79me1 by gene length (Fig. 5e). Although H3K79me does not play any role in histone exchange (Fig. 2c–e), multiple H3K79 methylation can be used to trace the frequency of Dot1p occupation on chromatin, because Dot1p is a nonprocessive enzyme. As shown in Fig. 5e (left panel), the long genes showed higher H3K79me3 levels than those of shorter lengths (both 1–2 kb and 0.5–1 kb). On the other hand, the 0.5–1 kb gene group had a higher level of H3K79me1 (Fig. 5e, right panel). These data suggest that Dot1p more frequently approaches long genes to facilitate histone exchange. When we used ChIP-seq to consider the levels of histone H3, we found that Dot1p had a stronger influence on long genes (Fig. 5f). A boxplot of the $\log_2$ difference ratio of histone H3 in $dot1\Delta$ versus wild-type cells in terms of gene length showed that the levels of histone H3 were decreased in long genes. Together, these data show that the effect of Dot1p on histone exchange is stronger for long genes, which have a higher frequency of contact between Dot1p and chromatin. Moreover, this result is consistent with the widely accepted notion that histone exchange is a major factor in chromatin dynamics.

**Dot1p regulates chromatin accessibility**. To confirm that Dot1p modulates the dynamic changes of chromatin, we performed MNase-seq and ATAC-seq (Assay for Transposase-Accessible Chromatin with high-throughput sequencing) analyses using the relevant deletion mutants. If Dot1p regulates the dynamics of chromatin structure, changes in nucleosome positioning or chromatin accessibility may be observed in $dot1\Delta$ cells. We performed MNase-seq to assess the effect on nucleosome positioning and nucleosome occupancy and ATAC-seq, a technique with a similar rationale to that of DNase-seq, to assess the effect on chromatin accessibility. However, the analysis of MNase-seq showed that there were no significant change in nucleosome phasing (Supplementary Fig. 6a), occupancy or positioning in $dot1\Delta$ cells compared to wild-type cells (Supplementary Fig. 6b). These data indicate that Dot1p does not globally change nucleosome positioning.

Next, nucleosome accessibility was measured by mapping the fragment less than 100 bp among the read counts of ATAC-seq[43]. Since the loss of H4K16ac is well known to trigger the formation of condensed chromatin[26, 44–46], we first confirmed the change of chromatin accessibility in $sas2\Delta$ cells. As expected, accessibility was significantly and globally decreased in $sas2\Delta$ cells compared to wild-type cells (Fig. 6a). Interestingly, in $dot1\Delta$ cells compared to wild-type cells, accessibility was reduced on the transcribed regions of genes but only slightly increased at their 3′-ends (Fig. 6a, middle). Thus, compared to the chromatin of wild-type cells, that of $dot1\Delta$ cells is less accessible through the transcribed regions of genes (Fig. 6a, center panel). Chromatin accessibility was increased in $set2\Delta$ cells relative to wild-type cells (Fig. 6a, right panel), which may reflect that the transcribed regions of genes have hyper-acetylated chromatin in $set2\Delta$ mutant cells[4, 5, 47, 48]. We also confirmed our heatmap analysis using boxplots of $\log_2$ fold change values on transcribed regions (Fig. 6b). As shown in Fig. 6b, $sas2\Delta$ and $dot1\Delta$ cells showed global reductions in nucleosome accessibility ($\log_2$ fold change <0), whereas $set2\Delta$ cells displayed increased accessibility ($\log_2$ fold change >0).

From all the annotated genes of *Saccharomyces cerevisiae* ($n = 6692$), we separated the genes of each mutant based on their increased or decreased accessibility compared to wild-type cells (Fig. 6c). Our stacked percent graph showed that most of the genes in $sas2\Delta$ cells exhibited decreased chromatin accessibility relative to wild-type cells ($n = 5585$). About 50% of the total genes showed decreased accessibility in $dot1\Delta$ cells ($n = 3287$) and such

difference between $sas2\Delta$ and $dot1\Delta$ cells is not due to a change in increased accessibility ($n = 340$), but rather the increased steady genes ($n = 3065$). In $set2\Delta$ cells, 2580 genes showed increased accessibility; although this accounted for less than 50% of the total gene set, we note that accessibility was increased in most of the genes if we excluded the steady genes ($n = 3467$). These results are important as they suggest that: (1) the H4K16ac–Dot1p pathway functions to increase chromatin accessibility; and (2) Dot1p modulates nucleosome accessibility, perhaps via nucleosome assembly activity at the transcribed regions of genes. Therefore, we envision that Sas2p and its downstream partner, Dot1p, regulate nucleosome accessibility during transcription, and that Dot1p further uses its histone-chaperoning and histone methyltransferase activities to assemble epigenetic information on chromatin, thereby modulating later processes, such as DNA replication, recombination, and repair.

**Discussion**

The findings of this study indicate that the histone H3K79 methyltransferase, Dot1p, is a histone chaperone and plays a pivotal role in the histone exchange that balances nucleosome dynamics during transcription (Fig. 6d). We present a number of important findings that link Dot1p to genome/chromatin stability. First, we report a crosstalk between H4K16ac and H3K79me, and show that it is central for the distribution of Dot1p-mediated H3K79me on the transcribed regions of genes. Second, the loss of *DOT1* suppresses histone exchange on the transcribed regions and down-regulates cryptic transcripts in $set2\Delta$ cells. Third, Dot1p has a nucleosome assembly activity in vitro. Fourth, Dot1p is able to stimulate the nucleosome-sliding activities of ATP-dependent chromatin remodelers, such as Chd1p. Fifth, the functions of Dot1p in nucleosome assembly and histone exchange are independent of histone H3K79me. Finally, Dot1p itself can affect global chromatin dynamics by altering nucleosome accessibility at the transcribed regions of genes.

We found that the effects of Dot1p on nucleosome assembly and histone exchange were more pronounced in long genes (>2 kb), reminiscent of the higher histone exchange in long genes in Set2p-depleted cells[1, 4]. These findings suggest that long genes are more sensitive to alteration of chromatin dynamics and to the functions of Dot1p. In addition, histone turnover and histone chaperone activities has been shown to be important in the context of DNA damage situation[49]. We thus propose that Dot1p has dual methylation-dependent and methylation-independent functions. The methylation-independent function of Dot1p involves the facilitation of histone turnover, which balances chromatin dynamics and consequently helps the cell to maintain chromatin stability. The methylation-independent function of Dot1p includes its conventional H3K79me methyltransferase activity. Given that H3K79me is broadly involved in the DNA damage response, it may be described as a 'contingency platform' for a DNA damage situation[50] especially in longer genes, adding another layer of chromatin protection.

The correlation between relative histone exchange (replication-independent histone turnover) and H3K79me3 level has been reported, such that genes with high turnover display low H3K79me3 levels. This indicates histone exchange is likely to be an alternative means to reduce H3K79me[51], rather than the activity of an as yet unidentified H3K79 demethylase. Moreover, loss of Rtt109p, a histone H3K56 acetyltransferase that is involved in the exchange of newly synthesized histones, was reported to increase the ratio of H3K79me3/H3K79me1 near promoters[52]. These results suggest that the assembly, deposition, and exchange of new histones prominently reduce H3K79me to maintain its chromatin concentration[53]. Consistent with this hypothesis, we

herein show that Dot1p acts as a chaperone for histone exchange, consequently maintaining the level of H3K79me.

Taken together, our results show that the histone methyltransferase, Dot1p, functions as a histone chaperone to directly regulate chromatin dynamics. Since histone exchange is also important for transcription elongation and enhancer functions in mammals, our result may provide a paradigm for understanding the functions of the human ortholog, DOT1L, in transcription regulation[54, 55]. Furthermore, our findings may contribute to the development of future therapeutic interventions since a number of diseases, including MLL, have been associated with functional alterations of DOT1L[56, 57].

## Methods

**Yeast strains**. The yeast strains used in this study are listed in Supplementary Table 1. Except for the exchange-assay strain, all yeast strains were grown in YP medium containing 2% glucose. Cultures were incubated at 30 °C until the $OD_{600}$ reached 0.8.

**TCA precipitation and western blotting**. Total proteins were prepared with trichloroacetic acid (TCA) precipitation. Cells were grown in 50 ml YP medium containing 2% glucose, and harvested at $OD_{600}$ of 0.8. The cells were suspended in 20% TCA before undergoing cell lysis with glass bead beating. Pellets were collected and washed twice with 5% TCA. The pellets with 5% TCA solution were incubated on ice for 30 min. After removing supernatant, pellets were suspended in SDS sample buffer (60 mM Tris-HCl, pH 6.8, 2% SDS, 4% β-mercaptoethanol, 20% glycerol) containing bromophenol blue (1 mg ml$^{-1}$). The protein samples were then subjected to western blotting. The protein samples were resolved by 15% sodium dodecyl sulfate–polyacrylamide gel electrophoresis (SDS-PAGE) gel and transferred to nitrocellulose membrane at constant voltage 10 V for 1 h. After rinsing the membranes in Ponceau S solution to check transfer quality, the membranes went through blocking reaction with tris-buffered saline and Tween-20 (TBS-T) buffer (10 mM Tris-HCl, pH 7.5, 3 mM KCl, 150 mM NaCl, 0.1% Tween-20) containing 5% of skim milk for 1 h. The membranes were then incubated in TBS-T buffer containing 5% skim milk with the relevant primary antibodies (see Supplementary Table 2) for 1 h at room temperature. The membranes were washed with TBS-T buffer three times, 5 min each, and were incubated with rabbit secondary antibody (Jackson Lab; 111-035-003). After 45 min of incubation with the secondary antibody in TBS-T buffer containing 5% skim milk, the membranes were washed with TBS-T for four times, 5 min each. The blotted membranes were applied with Immobilon western chemiluminescent HRP substrate (Millipore) to detect blot band in X-ray film. Uncropped scans of the western blot analyses are shown in Supplementary Fig. 10.

**Peptide pull-down assay**. Dot1p was expressed from pET21a-TEV-MBP and purified. The MBP tag was removed by TEV cleavage reaction. The biotinylated N-terminal H4 tail (amino acids 1–21) fragment was bound to streptavidin-coated Dynabeads M280 (Thermo Fisher Scientific; 11205D) in coupling buffer (25 mM Tris-HCl, pH 8.0, 1 M NaCl, 1 mM dithiothreitol (DTT), 5% glycerol, 0.03% NP-40) at 4 °C for overnight. Purified Dot1 and the bead-bound H4 tail peptides were mixed in peptide-binding buffer (25 mM Tris-HCl, pH 8.0, 150 mM NaCl, 1 mM DTT, 5% glycerol, 0.5% Triton X-100) and incubated at 4 °C for 1 h. After washing five times with wash buffer (peptide-binding buffer containing 200 mM NaCl), the samples were analyzed by SDS-PAGE and subjected to western blotting.

**GST-protein pull-down assay**. The glutathione S-transferase (GST) fusion proteins were expressed in *Escherichia coli* (Rosetta2, Novagen) at 18 °C for 10 h and were purified using glutathione superflow resin (Clontech). Then, 5 μg of GST fusion proteins and 10 μg of histone dimer, tetramer, and octamer were mixed in 200 μl of PDB buffer (50 mM HEPES-KOH, pH 7.4, 150 mM NaCl, 10% glycerol, 0.5 mM DTT, 1% Triton X-100, and 1 mM phenylmethylsulfonyl fluoride) and incubated at 4 °C for 2 h. After washing resins with PDB buffer three times, resins were boiled in SDS sample buffer for 5 min. The protein samples were resolved by SDS-PAGE gel, and analyzed by Coomasie blue staining. The uncropped scans of coomasie blue stained SDS-PAGE gel is shown in Supplementary Fig. 11.

**$K_D$ calculation**. The equilibrium dissociation constants ($K_D$) were measured by a BLItz system (ForteBio, Inc.). The streptavidin biosensor dip was soaked in peptide-binding buffer for 10 min before measurement steps. We used peptide-binding buffer as the BLItz assay buffer, both to soak the streptavidin biosensor dip and to perform between-step washes. Biolayer interferometry assays consisted of five steps: initial base line (30 s), loading, base line (30 s), association, and dissociation. Biotinylated H4 (1–20) peptides were immobilized on streptavidin biosensor dip. For the loading step, protein concentrations were adjusted to yield a signal intensity in the range of 1 to 2 nm, thereby ensuring that the sensors were not saturated. Times and protein concentrations for the association and

dissociation steps were as indicated in the figures and legends. Control values, measured using empty (no protein loaded) sensors, were subtracted from experimental values before data processing. The analytic program was set to the global mode to a 1:1 binding model by the BLItz Pro program, from which the equilibrium constant ($K_D$) and association ($K_a$) and dissociation ($K_d$) rate constants were calculated.

**ChIP-seq**. Yeast strains were grown in YP medium containing 2% glucose (YPD). Cultures were incubated at 30 °C until the $OD_{600}$ reached 0.8. The cells were then subjected to 1% formaldehyde cross-linking, sonication, and immunoprecipitation with the relevant antibodies (listed in Supplementary Table 2). For replication-independent histone modification ChIP-seq, the yeast strains were grown in YPD to $OD_{600}$ 0.5. Then, 20 μl of alpha factor (10 mg ml$^{-1}$) was added to 100 ml of medium and incubated for 2 h at 30 °C. In the case of the histone exchange strain, cells were grown in YP medium containing 2% raffinose. At $OD_{600}$ 0.5, cells were arrested in $G_1$ phase by adding 20 μl of alpha factor (10 mg ml$^{-1}$) to 100 ml of medium, and incubated the mixture for 4 h at 30 °C. Galactose was added (final concentration, 2%), and the cells were incubated for 1 h, and then cross-linked with 1% formaldehyde. To remove RNA and to decrosslink proteins, the Immunoprecipitates was treated with 50 μg of RNase A for 1 h at 37 °C, and proteinase K (Sigma) for 2 h at 55 °C. After an overnight incubation at 65 °C, immunoprecipitates were recovered, and the DNA was purified using a Qiagen PCR purification kit. ChIP-seq libraries for genome-wide sequencing were prepared using a NEXTflex ChIP-Seq kit (Bioo Scientific; cat. no. 5143). The prepared libraries were sequenced on an Illumima HiSeq2500 using the single-end method (50 bp reads) The ChIP-seq reads were mapped onto the sacCer3 reference genome using the Bowtie2 align program, and data normalization analysis was performed using the MACS2 peak calling program. Log$_2$ values between mutant and wild-type cells were calculated using bamCompare in the Deeptools, a data analysis program for high-throughput sequencing. The read counts and RPKM values were identified using the bedtools and bwtool utility program. RPKM values were calculated as: RPKM = (number of reads mapped to a gene × 1E + 09)/ (length of the gene × number of total mapped read counts in the experiment). The scatter plots and $R^2$ values showing the duplicated ChIP-seq sample variation were included in Supplementary Figs. 7, 9a, and b.

**ChIP-seq spike-in normalization**. For quantitative analysis of our H4K16ac and H3 ChIP-seq data, we normalized the mutant/wild-type IP ratio with respect to a spike-in control[58], which should (in principle) be constant in all ChIP samples. Before the cell lysis step of the ChIP process, we mixed two types of formaldehyde-fixed cells (*S. cerevisiae*:*S. pombe* = 5:1; $OD_{600}$ ratio). We then subjected this mixture to the ChIP-seq library preparation process. For computational analysis, the normalized ratio of IP/input ($E^{IP}$) was calculated as:

$$E^{IP} = (\text{read count of IP/input mapped to the } S.\ cerevisiae \text{ genome})$$
$$/(\text{read count of IP/input mapped to the } S.\ pombe \text{ genome})$$

The normalization factor ($N$) for $IP^{mutant}$: $IP^{WT}$ was calculated as:

$$N \times \text{read count of } IP^{mutant} : \text{read count of } IP^{WT} = E^{IP} \text{ of mutant} : E^{IP} \text{ of wild} - \text{type}.$$

We used the normalization option '-scaleFactors' in the Deeptools bamCompare tool, and the normalization factor ($N$) was used as '-scaleFactors N:1' for normalization of the mutant over wild type.

**mRNA-seq**. Total RNA was prepared by the hot-phenol method. mRNA isolation and mRNA-seq library preparation were performed using the NEXTflex Rapid Directional mRNA-Seq Bundle (Bioo Scientific; cat. no. 5138-10). mRNA library sequencing was performed on a HiSeq2500 using the single-end method (50 bp reads), and the raw reads were aligned to the sacCer3 genome using the STAR aligner. The number of mapped reads was further analyzed using Cuffdiff[59]. The scatter plots and $R^2$ values showing the sample variation of the mRNA-seq duplicates were included in Supplementary Fig. 8.

**MNase-seq**. For preparation of MNase-seq library, formaldehyde-fixed cell pellets were resuspended in 10 ml of pre-incubation buffer (20 mM EDTA, 0.7 M mercaptoethanol), washed, and resuspended in 2 ml of zymolyase buffer (1 M sorbitol, 50 mM Tris-Cl, pH 8.0, 5 mM mercaptoethanol) containing 2.5 mg ml$^{-1}$ zymolyase (20T; US Biological). Lysed cells were spheroplasted at 30 °C for 20 min and chromatin was isolated. Pelleted nuclei were digested with 200 units of MNase (NEB) at 37 °C for 20 min (confirmed as a typical digestion condition by comparison of fragment sizes), and then reverse cross-linked in 10 mM EDTA and 1% SDS buffer with proteinase K for 6 h at 37 °C. The DNA was extracted with phenol/chloroform, purified by ethanol precipitation, and resolved by 2% agarose gel electrophoresis in 1xTAE (50x TAE: 242g Tris base, 57.1ml acetic acid, and 100mL of 500mM EDTA (pH 8.0)) buffer. Mononucleosomal fragments were recovered

with a gel extraction kit (Qiagen). MNase libraries were prepared using a TruSeq Nano DNA Library Prep Kit (Illumina; FC-121-4001), and sequenced using a HiSeq2500 with the paired-end method. The obtained MNase-seq reads were aligned using Bowtie2, and nucleosome positions and occupancies were identified by DANPOS2. The average plot was graphed with the DANPOS2 profiler. The heatmap was clustered using Cluster 3.0 with 3 $k$-means and profiled using Treeview. The scatter plots and $R^2$ values showing the sample variation of MNase-seq duplicates were included in Supplementary Fig. 9d.

**ATAC-seq.** Yeast cells (2.5 million cells) were harvested at O.D. 0.6~0.8 ml$^{-1}$ (mid-log phase), and spheroplasted in 200 μl of Sorbitol buffer (1.4 M Sorbitol, 40 mM Tris-HCl, pH 7.5, 0.5 mM MgCl$_2$) with 10 μl of 50 mg ml$^{-1}$ of zymolyase (20T; US Biological). After washing twice with 100 μl of ice-cold Sorbitol buffer, the spheroplasts were incubated in fresh TD buffer (20 mM Tris-HCl, pH 8.0, 10 mM MgCl$_2$, 20% dimethylformamide) with transposase at 37 °C for 4 h. QIAquick PCR purification kit was used to purify DNA fragments, and transposition reactions were performed[43, 60]. The transposition reaction was incubated in 37 °C water bath for 4 h in 23.75 μl of TD buffer containing 1.25 μl of Tn5 transposase. We used HiFi HotStart ReadyMix (KAPA; KK2601) for library amplification following provided manual except number of PCR cycle: we performed 14 cycle for PCR amplification. The amplified library was purified with a QIAquick PCR purification kit (Qiagen; cat. no. 28106) and sequenced using a HiSeq2500 system with the paired-end method. The ATAC-seq data were analyzed as described in the ChIP-seq section. The scatter plots and $R^2$ values showing the same variation of ATAC-seq duplicates were included in Supplementary Fig. 9c.

**Nucleosome assembly assay.** For nucleosome assembly assay, 147 bp DNA fragments were amplified from pGEM-3z/601 with the primers listed in Supplementary Table 3. *Xenpous laevis* core histones (200 ng) were incubated with Dot1, Dot1(101−140Δ), (1.5, 2.5, 3.2 nM), or GST-Nap1p (2 μg) at 30 °C for 30 min, and then 200 ng of the 147 bp DNA was added to each reaction. After incubation for 90 min at 30 °C, 500 ng of pRS405 DNA plasmids were mixed into each reaction, using 5 × Reaction buffer (100 mM Tris-HCl pH 8.0, 0.5 mM EDTA, 5 mM DTT, and 50% glycerol), and incubation was continued for 15 min at 30 °C. The samples were then resolved by electrophoresis on a nondenaturing 5% (w/v) polyacrylamide TBE gel. Mononucleosomes were stained with EtBr, and western blotting analysis was performed with an antibody raised against histone H3 (Millipore; 05-928). The Uncropped scans of the western blot is shown in Supplementary Fig. 10, and the uncropped and pre-inversion image of EtBr stained native polyacrylamide gel is shown in Supplementary Fig. 11.

**Gel-shift assay for remodeler activity.** Short oligonucleosomes were reconstituted through an octamer-transfer method using native HeLa nucleosomes and $^{32}$P-labeled 216 bp 601-DNA probe. DNA templates were PCR amplified from the pGEM-3z/601 plasmid using specific primers (see Supplementary Table 3). For the sliding assays, reconstituted nucleosomes were preincubated with an increasing amount of wild-type or mutant Dot1 proteins (25, 50, and 75 ng for Fig. 4c; 25 and 50 ng for Supplementary Fig. 6) for 30 min at 30 °C in remodeling buffer (10 mM Tris-HCl, pH 7.4, 5% glycerol, 100 μg ml$^{-1}$ bovine serum albumin, 1 mM DTT, 0.02% NP-40, 40 mM NaCl, 2.5 mM MgCl$_2$, 2.5 mM ATP) and then mixed with chromatin remodelers and incubated for an additional 45 min at 30 °C. In all cases, a suitable concentration of chromatin remodeler was determined by prior titration. The reactions were stopped by the addition of 1 μg of *Hin*dIII-digested DNA and 1 μg of long-oligo-nucleosomes. The mixtures were resolved by 5% native polyacrylamide gel electrophoresis, and the gels were dried and exposed using a FLA-7000 (Fuji). The uncropped image of exposed native polyacrylamide gel is shown in Supplementary Fig. 11.

**Spotting assay.** When the OD$_{600}$ indicated that the yeast had reached stationary phase, equivalent amounts of cells (OD$_{600}$ 0.5) were serially diluted fivefold and spotted onto SC (synthetic complete) and SC-His+Gal plates. The plates were incubated at 30 °C for 2−3 days, and photographs were obtained when the yeast growth reached stationary phase.

**Data availability.** The RNA-seq, MNase-seq, ATAC-seq, and ChIP-seq datasets have been deposited to the GEO under accession number GSE106450.

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

## Acknowledgements

We thank Drs. E.S. Choi and J.L. Workman for critical readings and discussions. We thank Dr. K.S. Kang and T. Kim for critical advice and discussions regarding the computational analysis. We wish to acknowledge Drs. L. Pillus and J. Kim for kindly providing the constructs. This research was supported by the Basic Science Research Program through the National Research Foundation of Korea (NRF) funded by the Ministry of Science and ICT (2016R1A2B2006354). This work was also supported by grants from the KAIST Future Systems Healthcare Project funded by the Ministry of Science and ICT.

## Author contributions

S.L. and D.L. formulated the concepts and designed the experiments. S.L. performed most of the experiments. H.J. performed the MNase-seq preparation and assisted in the research. S.O. and K.J. performed the gel-shift assays and assisted in the research. Y.C., M.K., H.D.S., C.S.K., and J.C. commented on the manuscript. S.L. wrote the manuscript under the technical supervision and mentorship of D.L.

## Additional information

**Competing interests:** The authors declare no competing financial interests.

