## [Peer Review File · Nature Communications]

Reviewers' comments:

Reviewer #1 (Remarks to the Author):

In this paper, Lee et al. describe an uncharacterized and non-enzymatic role of the histone H3 lysine-79 methyltransferase, Dot1. Although the methyltransferase activity of Dot1 has been well characterized for nearly two decades, this manuscript provides strong evidence for methyltransferase-independent functions of Dot1 including in vitro nucleosome assembly activity and stimulating ATP-dependent chromatin remodeling. The authors also show that H4K16ac is a key factor in the distribution of Dot1 mediated H3K79 methylation and that the loss of Dot1 can suppress cryptic transcription in *set2Δ* cells.

The results presented in the paper are rigorous and shed new light on the role of Dot1 in transcriptional regulation. As non-enzymatic roles of several other methyltransferases are also now being elucidated, this work seems very timely. Thus, this paper will be of interest to the chromatin and transcription community.

Before publication, however, we have several suggestions listed below that should be addressed:

1. Figure 3b-d: If Sas2-mediated H4K16ac is responsible for the distribution of Dot1 on transcribed regions, what effect would a *sas2Δ* or a *set2Δsas2Δ* have on the levels of H3 and on histone exchange in general?
2. Figure 4: In the chromatin sliding assay, can you be certain that the ability of Dot1 to stimulate Chd1 isn't a general function of other nucleosome-binding enzymes? The 101-140 region of Dot1 is clearly necessary for its stimulatory function but is it sufficient?
3. "long gene" sequencing analysis: Have the necessary measures been taken to ensure that gene length bias hasn't been introduced into the analysis?

Reviewer #2 (Remarks to the Author):

Soyun Lee et al have investigated the role of Dot1 in regulating nucleosome dynamics. They show that a cross talk between H4K16Ac and H3K79me and show that H4K16Ac regulates the Dot1p mediated distribution of H3K79 methylation on euchromatin. They show loss of Dot1 suppresses histone exchange and cryptic transcription of *set2Δ* cells in a methylation –independent manner. They show Dot1 has nucleosome assembly activity and enhances chromatin remodeler activities, including that of Asf1p. They demonstrate that Dot1 preferentially affects histone exchange in long genes. Lastly they show that Dot1 regulates chromatin accessibility, especially on transcribed regions of genes.

General comments:

The manuscript presents a number of novel and important advances in our understanding of the methyltransferase, Dot1p. The manuscript is very well written and the conclusions drawn are sound and based on high quality data.

As a general comment it is difficult to see differences in the peak values in *set2* and *set2Δ dot1Δ* double mutants in the way the data has been presented here. Perhaps a close-up of the peaks showing a modulated response in the *set2 dot1* double might be more appropriate?

Minor comments:

201: We used (the) a differential expression program

209: Set2-dep(e)leted

Figure 1: It is not clear from the text or figure legends that these different histone mutations are being overexpressed. This should be made clearer in both cases.

Figure 2: Need dot1 and set2 single mutant controls (either here or in supplementary).

References: Reference manager has failed to abbreviate and capitalize many journal names, eg Molecular cell should be Mol. Cell etc.

RPKM is not clearly defined.

The rationale for using MNase-seq and ATAC-seq, and what question being addressed here is not clearly explained. They should not assume that the reader is familiar with these techniques.

Reviewer #3 (Remarks to the Author):

In this manuscript Lee et al report that *S. cerevisiae* Dot1 regulates nucleosome dynamics through a histone chaperone activity that is independent of its previously well-characterised H3K79me activity. This is an important finding given that Dot1 is evolutionarily conserved in humans and its dysregulation is associated with MLL-leukemia. It also raises the possibility that other histone methyltransferases may function in a similar manner. This is of interest because methylation-independent roles have been suggested for another KMTase (*S. pombe* Set1). The authors show that H4K16ac is important for Dot1 recruitment to chromatin, that loss of Dot1 suppresses cryptic transcription in *set2Δ* cells, that Dot1 promotes nucleosome assembly in vitro and that Dot1 promotes histone exchange independently of its KMTase activity. The conclusions of the study are generally supported by the data, however there are some points which need to be addressed.

1. The number of biological repeats used to generate the ChIP-seq, RNA-seq, MNase-seq and ATAC-seq datasets is not specified (as far as I can tell). It should be, because single experiments are not acceptable. This is particularly important as the impact of Dot1 deletion on cryptic transcript levels, histone occupancy and exchange are relatively modest. Some attempt to show the levels of variability in these experiments should also be included.

2. There is a 'disconnect' between the data showing that H4K16ac is important for Dot1 recruitment and the failure of *sas2Δ* to suppress the cryptic transcript phenotype of *set2Δ*. If H4K61ac is really critical for Dot1 recruitment then deletion of SET2 should suppress the cryptic transcription phenotype of *set2Δ*, but it does not. This should be addressed/discussed. The supplementary information also shows that H4 tail binding is not necessary to potentiate the action of ATP-dep remodelers.

3. It has previously been shown that loss of Asf1 suppress the increased histone exchange that occurs in *set2Δ* cells. The author's indicate that that loss of Dot1 does the same. Therefore for Fig 5 it would have been very useful to have also compared histone turnover in *set2Δ* and *set2Δ dot1Δ* cells using the FLAG/Myc approach (rather than just measuring H4K16ac as in Fig 3).

4. Throughout the manuscript there is a lack of relevant detail in the Figure legends. For example the heat maps are poorly described and at present these figures add little to the study. The number of repeats should be specified for all experiments. What does the arrow head in Fig1a specify? What type of gel is shown in Fig 4a?

5. The materials and methods section lacks a description of the GST-pulldown experiment (Fig 4a). Here the reference to histone octamers (line 302) is probably misleading as octamers are only

stable in the presence of DNA. Lines 309 and 310 also suggest that both Asf1 and Nap1 were used as controls in this experiment. Only the data relating to Nap1 is shown.

6. Fig 2a. The authors indicate that "sas2 Δ set2 Δ cells displayed a slight decrease in growth compared to set2 Δ " I don't think that this conclusion can be made on the basis of the data shown in this Fig.

7. In the discussion the authors suggest that Dot1 may participate in chromatin dynamics in subtelomeric regions. Why did they not analyse these regions in their genome-wide experiments?

Minor Points

1. Line 32. Change "cryptic transcriptions" to "cryptic transcription"
2. Line 40 Change "The chromatin structure" to "Chromatin structure"
3. Lines 183-184. I don't understand the use of "stabilized" in this context. These lines should be reworded.
4. Lines 472-474. This sentences reads as if loss of Dot1 up-regulates cryptic transcription in set2 Δ cells, when in fact the opposite is true.
5. Lines 476. Asf1 is not an ATP-dep remodeler. I think the authors mean Chd1 here
6. Lines 500-501. The sense of this sentence is not clear, please reword.
7. Line 207 "DOT1" should be italicized
8. Line 223 replace "translation" with "transcription"

Point-by-point response to reviewer's comments

Reviewer #1 (Remarks to the Author):

In this paper, Lee et al. describe an uncharacterized and non-enzymatic role of the histone H3 lysine-79 methyltransferase, Dot1. Although the methyltransferase activity of Dot1 has been well characterized for nearly two decades, this manuscript provides strong evidence for methyltransferase-independent functions of Dot1 including in vitro nucleosome assembly activity and stimulating ATP-dependent chromatin remodeling. The authors also show that H4K16ac is a key factor in the distribution of Dot1 mediated H3K79 methylation and that the loss of Dot1 can suppress cryptic transcription in set2Δ cells.

The results presented in the paper are rigorous and shed new light on the role of Dot1 in transcriptional regulation. As non-enzymatic roles of several other methyltransferases are also now being elucidated, this work seems very timely. Thus, this paper will be of interest to the chromatin and transcription community.

Before publication, however, we have several suggestions listed below that should be addressed:

1. Fig. 3b-d: If Sas2-mediated H4K16ac is responsible for the distribution of Dot1 on transcribed regions, what effect would a sas2Δ or a set2Δsas2Δ have on the levels of H3 and on histone exchange in general?

Our comment: Thank you for your comment. We now have included the graphs of H4ac and H3 for *sas2Δ* and *sas2Δset2Δ* in **Supplementary Fig. 3**. The level of H4ac which is an indicative of the histone exchange do not show a significant difference in *sas2Δset2Δ* compared to *set2Δ* (**Supplementary Fig. 3b**). The level of H3 in *sas2Δ* was lower than wild-type cells (Supplementary Fig. 3c), and histone turnover measured by Flag-histone H3 in inducible strain displayed a slight decrease in *sas2Δ* (**Supplementary Fig. 3d**). These results support the approach of Dot1 by Sas2-mediated H4K16ac on transcribed region regulates histone exchange.

Although the unchanging level of histone exchange supports our previous observation of *sas2Δset2Δ* not suppressing cryptic transcription (**Fig. 2a**), it seems contradictory to the Sas2-mediated H4K16ac playing a role in the recruitment of Dot1 to the transcribed regions implicated by the decrease in H3K79me3 in *sas2Δ* cells (**Fig. 1a**). This is may be due to the difference in the chromatin context in wild-type and *set2Δ* cells. In *set2Δ* cells, the chromatins are hyper-acetylated as the HDAC enzymes are unable to be recruited via Set2-Rpd3S pathway. The hyper-acetylation state increases chromatin dynamics and increases chromatin accessibility, such that distributive enzymes like Dot1 can access the chromatin more easily. Moreover, the increased cryptic transcription in *set2Δ* cells may result in the recruitment of other HATs to the transcribed regions of cryptic transcription. We have shown the involvement of other HATs such as NuA4 at the promoter region at the promoters (**Fig. 1e**) and the emergence of cryptic promoters in the transcribed region of the genes in *set2Δ*

cells may therefore imply the involvement of other HAT in cryptic transcription situation. Taken together, the chromatin state in *set2Δ* cells are different from that of wild-type, such that Sas2-mediated H4K16ac may no longer necessary for the recruitment of Dot1.

2. *Fig. 4: In the chromatin sliding assay, can you be certain that the ability of Dot1 to stimulate Chd1 isn't a general function of other nucleosome-binding enzymes? The 101-140 region of Dot1 is clearly necessary for its stimulatory function but is it sufficient?*

Our comment: The chaperonic activity of Dot1 led to postulate that Dot1 may stimulate remodeling effect of remodelers, as histone chaperones like Asf1 has been shown to stimulate remodeling *in vitro*. As the reviewer suggested, it is possible that other chaperone proteins that have nucleosome binding activity stimulate chromatin remodeling of Chd1. Further characterization of other nucleosome binding proteins would be required to identify the role of nucleosome binding domain *per se*, in stimulation of Chd1.

As we mentioned in the line 256, it has been reported by Oh et al., (2010) that the Dot1 domain 101-140 plays a crucial role in the nucleosome binding of Dot1. In this paper, the authors characterized the domains of Dot1 by performing EMSA experiment using several truncation mutants including the *Dot1(101-140 Δ)* mutant. We tested all of these mutants for the stimulating role of Chd1 and only *Dot1(101-140 Δ)* mutant showed defect in the stimulation.

3. *“long gene” sequencing analysis: Have the necessary measures been taken to ensure that gene length bias hasn't been introduced into the analysis?*

Our comment: Thank you for raising an important issue. The gene length bias is one of the common bias that occur during analysis of high-throughput sequencing to produce metazoan average plot. We eliminated gene length bias by RPKM analysis in **Fig. 5d**. Usage of RPKM is commonly used in RNA-seq analysis, and is also used in ChIP-seq to remove gene length bias, by adopting the equation $RPKM = (\text{number of reads mapped to a gene} \times 1E+09) / (\text{length of the gene} \times \text{number of total mapped read counts in the experiment})$, which take gene length into consideration. We added the sentence “To eliminate analysis error due to variability of gene length and the effect of strong histone exchange near the promoters and 5'-ends of genes,” in line 385-387 to clarify the analysis method. Also, we would like to emphasize that the heatmap in **Fig. 5a** was sorted by gene length to prevent gene-length bias.

Reviewer #2 (Remarks to the Author):

Soyun Lee et al have investigated the role of Dot1 in regulating nucleosome dynamics. They show that a cross talk between H4K16Ac and H3K79me and show that H4K16Ac regulates

the Dot1p mediated distribution of H3K79 methylation on euchromatin. They show loss of Dot1 suppresses histone exchange and cryptic transcription of set2Δ cells in a methylation – independent manner. They show Dot1 has nucleosome assembly activity and enhances chromatin remodeler activities, including that of Asf1p. They demonstrate that Dot1 preferentially affects histone exchange in long genes. Lastly they show that Dot1 regulates chromatin accessibility, especially on transcribed regions of genes.

General comments:

The manuscript presents a number of novel and important advances in our understanding of the methyltransferase, Dot1p. The manuscript is very well written and the conclusions drawn are sound and based on high quality data.

As a general comment it is difficult to see differences in the peak values in set2 and set2Δ dot1Δ double mutants in the way the data has been presented here. Perhaps a close-up of the peaks showing a modulated response in the set2 dot1 double might be more appropriate?

Our comment: Thank you for your suggestion. We agreed with the reviewer's concern and revised **Fig. 2** and **Fig. 3** to contain a close-up of the peaks of mRNA-seq and log₂H3 data for set2Δ and set2Δdot1Δ mutants.

Minor comments

201: We used (the) a differential expression program

209: Set2-dep(e)leted

Our comment: Thank you. We revised the manuscript accordingly.

Fig. 1: It is not clear from the text or Fig. legends that these different histone mutations are being overexpressed. This should be made clearer in both cases.

Our comment: We apologize for not making it clear enough. For **Fig. 1**, we used wzy42 strains which the expression of histone H3 and H4 are maintained via *HHT-HHF2* plasmid. The histone mutant strains are not overexpressed strains but strains where *HHT-HHF2* plasmid is replaced with plasmids of respective histone mutations via FOA selection. We added the details in the legends and in the **Methods** sections.

Fig. 2: Need dot1 and set2 single mutant controls (either here or in supplementary).

Our comment: We included mRNA-seq results and the box-plot comparison of expression of single mutant *sas2Δ*, *dot1Δ* in **Supplementary Fig. S2**.

References: Reference manager has failed to abbreviate and capitalize many journal names, eg Molecular cell should be Mol. Cell etc.

Our comment: Thank you for letting us know. There was an error on ‘Term List’ in the reference program. We updated the program and fixed the error.

RPKM is not clearly defined.

Our comment: We apologize for not elaborating the term. We elaborated the term RPKM in the Line 139. We also added the equation for calculating RPKM in the analysis part of the **Materials and Methods** section.

The rationale for using MNase-seq and ATAC-seq, and what question being addressed here is not clearly explained. They should not assume that the reader is familiar with these techniques.

Our comment: Thank you for the constructive comment. We added the following sentences in Line 429 to further explain the rationale for using MNase-seq and ATAC-seq:

“If Dot1 regulates the dynamics of chromatin structure, changes in nucleosome positioning or chromatin accessibility may be observed in *dot1Δ* cells. We performed MNase-seq to assess the effect on nucleosome positing and nucleosome occupancy, and ATAC-seq, a technique that of similar rationale to DNase-seq, to assess the effect on chromatin accessibility.”

Reviewer #3 (Remarks to the Author):

*In this manuscript Lee et al report that *S. cerevisiae* Dot1 regulates nucleosome dynamics through a histone chaperone activity that is independent of its previously well-characterised H3K79me activity. This is an important finding given that Dot1 is evolutionarily conserved in humans and its dysregulation is associated with MLL-leukemia. It also raises the possibility that other histone methyltransferases may function in a similar manner. This is of interest because methylation-independent roles have been suggested for another KMTase (*S. pombe* Set1). The authors show that H4K16ac is important for Dot1 recruitment to chromatin, that loss of Dot1 suppresses cryptic transcription in *set2Δ* cells, that Dot1 promotes nucleosome*

assembly in vitro and that Dot1 promotes histone exchange independently of its KMTase activity. The conclusions of the study are generally supported by the data, however there are some points which need to be addressed.

1. The number of biological repeats used to generate the ChIP-seq, RNA-seq, MNase-seq and ATAC-seq datasets is not specified (as far as I can tell). It should be, because single experiments are not acceptable. This is particularly important as the impact of Dot1 deletion on cryptic transcript levels, histone occupancy and exchange are relatively modest. Some attempt to show the levels of variability in these experiments should also be included.

Our comment: We apologize for not specifying the number of biological repeats in the legends. The sentence specifying the number of repeats was added in the figure legends of the appropriate figures in the revised manuscript. We also included the scatter plot and R² value to show the sample variation of the repeat samples in **Supplementary Fig. 7-9**.

2. There is a 'disconnect' between the data showing that H4K16ac is important for Dot1 recruitment and the failure of sas2Δ to suppress the cryptic transcript phenotype of set2Δ. If H4K16ac is really critical for Dot1 recruitment then deletion of SET2 should suppress the cryptic transcription phenotype of set2Δ, but it does not. This should be addressed/discussed. The supplementary information also shows that H4 tail binding is not necessary to potentiate the action of ATP-dep remodelers.

Our comment: We appreciate your comment for pointing out an important issue. The phenotype of *sas2Δset2Δ* not suppressing the cryptic transcript phenotype of *set2Δ* was the point that puzzled us well. Fortunately, we managed come to a conclusion after checking the H4ac level in *sas2Δset2Δ*, which was the experiment that Reviewer 1 asked for. We have discussed in detail at the above section (Reviewer 1, comment 1) why *sas2Δset2Δ* would not suppress the cryptic phenotype of *set2Δ*. Briefly, the chromatin context in *set2Δ* cells are different to wild-type that the *set2Δ* chromatin are in its hyper-acetylation status and may stimulate non-specific recruitment Dot1, and may not necessitate Sas2 for Dot1 recruitment.

This hyper-acetylation status in *set2Δ* also helps to explain why the H4 tail binding domain of Dot1 (EDVDE domain) is not necessary for the stimulation of ATP-dependent remodelers. The chromatin accessibility increases when chromatin is hyper-acetylated and nucleosome binding proteins such as Dot1 may easily access to the transcribed region and facilitate histone exchange. This indicates that the histone exchange activity of Dot1 is mediated by the nucleosome binding activity but not the binding of histone H4 tail. In fact, histone H4 tail binding seems dispensable for the histone exchange activity *per se*. In the revised manuscript, we discussed **Supplementary Fig. 3** in detail to address this issue.

3. It has previously been shown that loss of *Asf1* suppress the increased histone exchange that occurs in *set2Δ* cells. The author's indicate that that loss of *Dot1* does the same. Therefore for Fig 5 it would have been very useful to have also compared histone turnover in *set2Δ* and *set2Δ dot1Δ* cells using the FLAG/Myc approach (rather than just measuring H4K16ac as in Fig 3).

Our comment: Thank you for your suggestion. As the reviewer suggested, we attempted FLAG/Myc approach in *set2Δ* and *set2Δdot1Δ* cells, and the result is presented as a box-plot in the figure below. The result of FLAG/Myc assay was a verification of our previous observation in H4ac: the turnover rate increased in *set2Δ*, and the increase was suppressed in *dot1Δset2Δ*. We fully agree that FLAG/Myc is an excellent approach to visualize the rate of histone exchange. However, we thought that H4ac data would represent more authentic phenotype than FLAG/Myc assay as FLAG/Myc assay includes media exchange from glucose to galactose while H4ac assay uses glucose media throughout. Moreover, since the data in **Fig. 3** proceeds the cryptic transcription data in **Fig. 2**, we thought using the same glucose medium would be a better choice. We therefore decided to present the FLAG/Myc assay only as reviewer's inspection.

4. Throughout the manuscript there is a lack of relevant detail in the Fig. legends. For example the heat maps are poorly described and at present these figures add little to the study. The number of repeats should be specified for all experiments. What does the arrow head in Fig1a specify? What type of gel is shown in Fig 4a?

Our comment: We again apologize for the legends. We added details in the figure legends such that it would contain all of the experimental conditions, symbols and scale bar. The arrow head in **Fig. 1A** was initially deployed for emphasis purpose but was removed in the revised manuscript as it did not have scientific significance. The type of gel shown in Fig. 4A was coomassie blue-stained gel. The **Materials and Methods** section and legends were changed accordingly.

5. The materials and methods section lacks a description of the GST-pulldown experiment (Fig 4a). Here the reference to histone octamers (line 302) is probably misleading as octamers are only stable in the presence of DNA. Lines 309 and 310 also suggest that both Asf1 and Nap1 were used as controls in this experiment. Only the data relating to Nap1 is shown.

Our comment: We apologize for the errors. The word ‘Asf1’ was removed. We described the GST-pulldown experiment in detail by adding a separate section in **Materials and Methods** section. The histone octamer was purified from YS14 core histone expression vector and stored a high salt buffer. We agree with the reviewer’s concern that the instability of octamers without presence of DNA as the octamers may not be stable under 150mM GST-pulldown experiment condition and disassemble into dimer forms. However, what we intended to test here was just the binding with the GST-proteins with octameric core-histone. We believe that the possible instability of the octamers would not affect our overall conclusion.

6. Fig 2a. The authors indicate that “sas2Δ set2Δ cells displayed a slight decrease in growth compared to set2Δ” I don’t think that this conclusion can be made on the basis of the data shown in this Fig.

Our comment: We agreed to the reviewer’s comment, and we revised the manuscript to describe that there was no change in the growth between *set2Δ* and *set2Δsas2Δ* cells. Also, we believe that the mRNA-seq is sufficient to demonstrate the change in cryptic transcription, and thus the spotting data in **Fig. 2a** was moved to **Supplementary Fig. 2a**.

7. In the discussion the authors suggest that Dot1 may participate in chromatin dynamics in subtelomeric regions. Why did they not analyse these regions in their genome-wide experiments?

Our comment: We removed sections regarding telomere silencing (Line 77-87, 498-505) as the sections were considered as deviations from our main findings. To answer the reviewer’s question, it is difficult to examine a precise role of Dot1 in transcribed regions in subtelomeric regions, because the subtelomeric regions contain wide intergenic regions and little long-length genes, making it difficult to directly analyze of the genes in this region (see

the figure below) Moreover, we believe the analysis of the genes in these regions is beyond the scope of this paper.

ChIP-seq data near *TEL06R* region of Chromosome VI. The figure represents the region from *TEL06R* to *YFR050C* (the distance is 21kb). The magenta box indicates the *TEL06R* region. The normalized ChIP-seq signal of H3K79me3 (blue), H3K79me1 (orange), and H4K16ac (purple) in wild-type cells is shown. Histone exchange in wild-type is represented as H4ac signal normalized by H3 (turquoise). In addition, red signal indicates histone turnover (red, \log_2 (Flag/Myc)) from Flag/Myc assay in wild-type cells.

Minor Points

1. Line 32. Change “cryptic transcriptions” to “cryptic transcription”
2. Line 40 Change “The chromatin structure” to “Chromatin structure”

Our comment: Thank you. We made the changes accordingly.

3. Lines 183-184. I don't understand the use of "stabilized" in this context. These lines should be reworded.

Our comment: We rephrased the Lines 183-184: "As transcription elongation is closely related to Set2-Rpd3S pathway which suppresses cryptic transcription, we examined the effects of Dot1p on cryptic transcription through Set2p."

4. Lines 472-474. This sentences reads as if loss of Dot1 up-regulates cryptic transcription in *set2Δ* cells, when in fact the opposite is true.

Our comment: Thank you for pointing it out. We rephrased the sentence: "Second, the loss of *DOT1* suppresses histone exchange on the transcribed regions of genes and down-regulates cryptic transcripts in *set2Δ* cells."

5. Lines 476. *Asf1* is not an ATP-dep remodeler. I think the authors mean *Chd1* here

Our comment: Thank you. We corrected it to *Chd1*.

6. Lines 500-501. The sense of this sentence is not clear, please reword.

Our comment: We removed Line 498-505 as we decided that the subject of telomere silencing deviates from our main story.

7. Line 207 "*DOT1*" should be italicized,

8. Line 223 replace "translation" with "transcription"

Our comment: We made the changes accordingly.

Again, we thank the reviewers for their time and effort reviewing our manuscript. The comments were constructive and, in addressing them, we greatly improved our insight into the histone chaperone activity of Dot1.

P. S. The following is a list of the major changes made to the revised manuscript:

1. To respond to a specific comment from Reviewer 3, we removed sections regarding telomere silencing (Line 77-87, 498-505) as the sections were considered as deviations from our main findings.
2. To respond to a general comment from Reviewer 2, we revised **Fig. 2** and **Fig. 3** to contain a close-up of the peaks of mRNA-seq and log₂H3 data for *set2Δ* and *set2Δdot1Δ* mutants.
3. To respond to a specific comment from Reviewer 2, we moved originally submitted **Fig. 2a** to **Supplementary Fig. 2a**, as we agreed to the reviewer's comment that *sas2Δset2Δ* does not suppress the cryptic phenotype of *set2Δ*.
4. To respond to a specific comment from Reviewer 2, we included mRNA-seq results and the box-plot comparison of the gene expression of *sas2Δ*, *dot1Δ* single mutants in **Supplementary Fig. 2**.
5. To respond to a specific comment from Reviewer 3, we performed FLAG/Myc experiment and showed the data as reviewer's inspection to verify our previous observation of the histone turnover suppression in *dot1Δset2Δ*.
6. To respond to a specific comment from Reviewer 3, we extensively discussed the issue of *sas2Δset2Δ* not suppressing the cryptic phenotype of *set2Δ* by discussing the data in **Supplementary Fig. 3**.
7. To respond to a specific comment from Reviewer 1, we included the graphs of H4ac *sas2Δ* and *sas2Δset2Δ* and the graph of H3 in *sas2Δ* in **Supplementary Fig. 3** to show the effect of the respective mutant on the levels of H3 and on histone exchange.
8. To respond to a specific comment from Reviewer 2, we revised the **Results** section to include a detailed rationale of using MNase-seq and ATAC-seq (Line 429)
9. To respond to a specific comment from Reviewers 1 and 2, we revised the **Results** section to include the definition of RPKM (Line 139) and revised the **Materials and Methods** section to include the equation used for the calculation of RPKM.
10. To respond to a specific comment from Reviewer 3, we revised the manuscript to describe that there was no change in the growth between *set2Δ* and *set2Δsas2Δ* cells which we previously argued for a marginal change. We therefore moved the previous **Fig. 2a** to **Supplementary Fig. 2a**.
11. To respond to a specific comment from Reviewer 2, we revised the figure legends and the **Materials and Methods** section to include the details of the *wzy42* strain system for analyzing histone mutant strains.

12. To respond to a specific comment from Reviewers 2 and 3, we revised the figure legends to include the details of the number of biological repeats, meaning of axis and colors for further clarity, and **Materials and Methods** section was revised accordingly.
13. To respond to a specific comment from Reviewer 3, we included scatter plots and R^2 values to show the sample variation of the repeat samples in newly added **Supplementary Fig. 7-9**.
14. To respond to a specific comment from Reviewer 3, we described the GST-pulldown experiment in detail by adding an independent section in **Materials and Methods** section.
15. To respond to a specific comment from Reviewer 3, we included the genome-wide H3K79me distribution and histone exchange rate as reviewer's inspection to show the difficulty of assessing the role of Dot1 in subtelomeric regions.
16. To respond to a specific comment from Reviewer 3, we revised the **Results** section to clarify the meaning of transcription elongation being stabilized by the Set2-Rpd3S pathway.
17. We removed *FLO8* data from originally submitted **Fig. 2b** due to space restriction as it was a redundant example among the four genes (*SPB4*, *STE11*, *PCAI*, *FLO8*).
18. We revised the **Abstract** section to reduce its length to meet the requirement of maximum 150 words.
19. We trimmed the **Introduction** and **Discussion** sections to reduce the number of words.

REVIEWERS' COMMENTS:

Reviewer #1 (Remarks to the Author):

The authors have done a good job in addressing our concerns. While the manuscript is significantly improved, a few notable issues should be addressed, and are listed below.

Major points:

- Lines 187-188: Authors state that "Deletion of DOT1 in the *rco1Δ* mutant background also decreased the mRNA RPKM compared to that in *set2Δ* cells, which exhibited cryptic transcription" (shown in Supplementary figure 2d). However, the authors do not provide the *rco1Δ* single deletion as a control for the genes measured in the figure. While it is conceivable that the effect seen in an *rco1Δ* is identical to a *set2Δ*, making a conclusion without the single deletion control is unsound. This issue needs to be resolved.
- Line 298-299: the explanation of the assay suggests all of the proteins are added together. Should be Nap1, Dot1, OR Dot1 Δ 101-140
- Line 304: How does the evidence that loss of Dot1 nucleosome binding disrupts assembly lead you to speculate that binding is a consequence of chaperone activity and not vice-versa? It could go either way, and this should be accounted for in the text.

Minor points (by line):

- 42: absence of [the] Set2
- 44: "increases" to "an increase in"
- 46: spell out H3K79me on first occasion
- 147: remove "the" from beginning of sentence
- 161-162: cells displayed, not showed, weak growth
- 269: *sas2 Δset2 Δ* [cells] compared to *set2 Δ* [cells]—"cells" frequently omitted after deletion throughout manuscript
- 271: "the chromatins are" should be "the chromatin is"
- 271: define the acronym "HDAC"
- 415: positioning not positing
- 471-472: "in the context of DNA damage" instead of "in the DNA damage situation"
- 896 and 901: indicating not indicting

Reviewer #2 (Remarks to the Author):

The authors have addressed my concerns. I have, however, found a number of minor typos and spelling errors that need to be addressed (the corrected form is below)

Line 42: The absence of the Set2-Rpd3S pathway makes chromatin more accessible to histone chaperones such as Asf1p, which results in the active assembly of histones and consequently increases histone exchange.

Line 119 (Fig. 1b, bottom), and there was a marked reduction in *sas2Δ* cells at transcribed regions but not near promoter regions.

Line 149: and hence increase the accessibility of chromatin, together with our finding that Dot1p is

Line 151 Dot1p may be involved in transcription elongation. During transcription elongation, the Set2-Rpd3S pathway suppresses cryptic transcription to prevent aberrant transcripts and to maintain normal transcription conditions.

Line 271 type and set2 Δ cells. When SET2 is lost, the chromatin is hyper-acetylated as the HDAC

Line 274 histone H3 in an inducible strain displayed

Line 277 cells is different from that of wild-type,

Line 278 may no longer be necessary for the recruitment of Dot1p.

Line 416 occupancy; and ATAC-seq, a technique with a similar rationale to that of DNase-seq, to assess the

Line 483 reduce H3K79me 53, rather than the activity of an as yet unidentified H3K79 demethylase.

Line 506 suspended in 20% TCA before undergoing cell lysis with glass bead beating.

Line 536 Inc.). The streptavidin biosensor dip was soaked

Line 546 analytic program was set to a global, 1:1 binding model (not sure if this is what you mean to say though).

Line 634 For nucleosome assembly assay, 147-bp DNA fragments were

Line 841 (Transcription End Site). For the heat maps, the y-axis indicates each gene and the x-axis indicates distance from TSS (Transcription Start Site)

Reviewer #3 (Remarks to the Author):

The authors have fully addressed all the points raised by this reviewer.

Point-by-point response

REVIEWERS' COMMENTS:

Reviewer #1 (Remarks to the Author):

The authors have done a good job in addressing our concerns. While the manuscript is significantly improved, a few notable issues should be addressed, and are listed below.

Major points:

· *Lines 187-188: Authors state that “Deletion of DOT1 in the rco1Δ mutant background also decreased the mRNA RPKM compared to that in set2Δ cells, which exhibited cryptic transcription” (shown in Supplementary figure 2d). However, the authors do not provide the rco1Δ single deletion as a control for the genes measured in the figure. While it is conceivable that the effect seen in an rco1Δ is identical to a set2Δ, making a conclusion without the single deletion control is unsound. This issue needs to be resolved.*

Our comment: Thank you for your comment. We agreed on the reviewer’s comment on the need for the rco1Δ single mutant. As we had to trim the main text to below 5000 words, we decided to remove the dot1Δ rco1Δ mRNA-seq data and also from the main text.

· *Line 298-299: the explanation of the assay suggests all of the proteins are added together. Should be Nap1, Dot1, OR Dot1 Δ101-140*

Our comment: Thank you for your comment. We corrected it.

· *Line 304: How does the evidence that loss of Dot1 nucleosome binding disrupts assembly lead you to speculate that binding is a consequence of chaperone activity and not vice-versa? It could go either way, and this should be accounted for in the text.*

Our comment: We agreed on the reviewer’s comment. We removed the sentence as we cannot exclude the possibility that the reduction of chaperone activity was consequent of the disruption of Dot1 binding on substrate.

Minor points (by line):

- 42: absence of [the] Set2
- 44: “increases” to “an increase in”
- 46: spell out H3K79me on first occasion
- 147: remove “the” from beginning of sentence
- 161-162: cells displayed, not showed, weak growth
- 269: *sas2* Δ *set2* Δ [cells] compared to *set2* Δ [cells]—“cells” frequently omitted after deletion throughout manuscript
- 271: “the chromatins are” should be “the chromatin is”
- 271: define the acronym “HDAC”
- 415: positioning not positing
- 471-472: “in the context of DNA damage” instead of “in the DNA damage situation”
- 896 and 901: indicating not indicting

Our comment: Thank you for your comment. We made the changes accordingly.

Reviewer #2 (Remarks to the Author):

The authors have addressed my concerns. I have, however, found a number of minor typos and spelling errors that need to be addressed (the corrected form is below)

Line 42: The absence of the Set2-Rpd3S pathway makes chromatin more accessible to histone chaperones such as Asf1p, which results in the active assembly of histones and consequently increases histone exchange.

*Line 119 (Fig. 1b, bottom), and there was a marked reduction in *sas2* Δ cells at transcribed regions but not near promoter regions.*

Line 149: and hence increase the accessibility of chromatin, together with our finding that Dot1p is

Line 151 Dot1p may be involved in transcription elongation. During transcription elongation, the Set2-Rpd3S pathway suppresses cryptic transcription to prevent aberrant transcripts and to maintain normal transcription conditions.

Line 271 type and set2Δ cells. When SET2 is lost, the chromatin is hyper-acetylated as the HDAC

Line 274 histone H3 in an inducible strain displayed

Line 277 cells is different from that of wild-type,

Line 278 may no longer be necessary for the recruitment of Dot1p.

Line 416 occupancy; and ATAC-seq, a technique with a similar rationale to that of DNase-seq, to assess the

Line 483 reduce H3K79me 53, rather than the activity of an as yet unidentified H3K79 demethylase.

Line 506 suspended in 20% TCA before undergoing cell lysis with glass bead beating.

Line 536 Inc.). The streptavidin biosensor dip was soaked

Line 546 analytic program was set to a global, 1:1 binding model (not sure if this is what you mean to say though).

Line 634 For nucleosome assembly assay, 147-bp DNA fragments were

Line 841 (Transcription End Site). For the heat maps, the y-axis indicates each gene and the x-axis indicates distance from TSS (Transcription Start Site)

Our comment: Thank you for your detailed comment. We revised the manuscript accordingly.

Reviewer #3 (Remarks to the Author):

The authors have fully addressed all the points raised by this reviewer.